# CAN AI TRULY REPRESENT YOUR VOICE IN DELIBERATIONS? A COMPREHENSIVE STUDY OF LARGE-SCALE OPINION AGGREGATION WITH LLMS

## ABSTRACT

Large-scale public deliberations generate thousands of free-form contributions that must be synthesized into representative and neutral summaries for policy use. While LLMs have been shown as a promising tool to generate summaries for large-scale deliberations, they also risk underrepresenting minority perspectives, raising fairness concerns in high-stakes contexts. Studying and fixing these issues requires a comprehensive evaluation at a large scale, yet current practice often relies on LLMs as judges, which show weak alignment with human judgments. We introduce DELIBERATIONBANK, a large-scale, human-grounded benchmark for deliberation summarization that contains (1) 3,000 participant-generated opinions across ten deliberation questions and (2) 4,500 human annotations evaluating summaries along four dimensions: representativeness, informativeness, neutrality, and policy approval. Using this benchmark, we train DELIBERATIONJUDGE, a domain-aligned evaluator that provides more reliable and efficient assessments than general-purpose LLM judges. With this evaluation framework, we benchmark 18 LLM summarizers and uncover consistent weaknesses, including systematic underrepresentation of minority viewpoints. Our benchmark and evaluator offer a scalable and reliable foundation for assessing deliberation summarization systems, supporting the development of more representative, equitable, and policy-relevant AI tools.[wcNP]

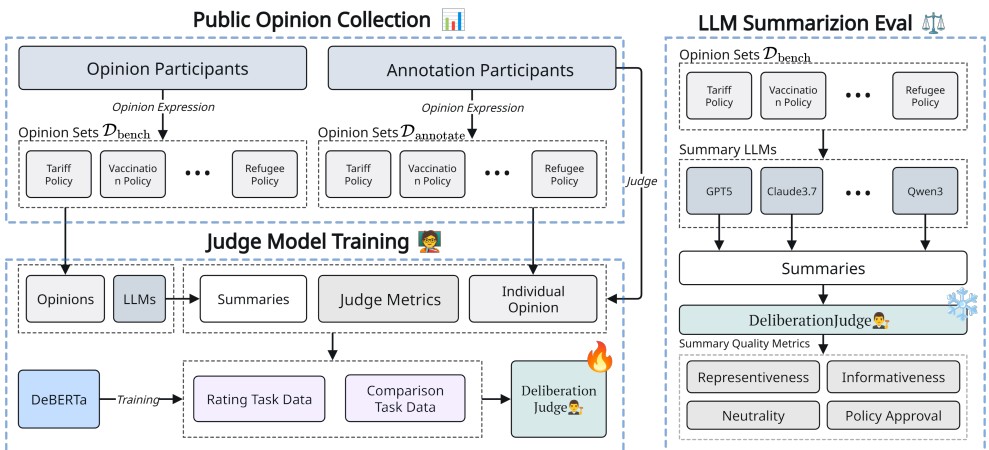

Figure 1: Overview of our benchmark framework(see §2), including opinion collection (§2.2), judge model training (§3), and LLM-based deliberation summarization evaluation (§2.1).[GiWi]

## 1 INTRODUCTION

Public and civic deliberations (e.g., citizens' assemblies and parliamentary debates) increasingly involve large numbers of free-form contributions from diverse participants. Summarization and reporting are essential yet costly components of these processes. In citizens' assemblies, facilitators must continuously synthesize perspectives during discussions and later produce comprehensive summaries for participants, stakeholders, policymakers, and the broader public (Landemore, 2020).

In parliamentary settings, members and their staff must maintain up-to-date representations of a complex, multi-stakeholder environment (Brandsma & Otjes, 2024). Turning hundreds of comments into decision-ready summaries requires more than compression: summaries must be informative and neutral while faithfully representing the full spectrum of viewpoints, including minority perspectives. At scale, ensuring efficiency without sacrificing representativeness and fairness remains a central challenge for public deliberation.

Classical computational pipelines such as clustering or dimensionality reduction can surface thematic structures (Chang et al., 2009; Roberts et al., 2014; Sievert & Shirley, 2014), but still require expert effort to interpret results and draft reports. Recent advances in LLMs offer a more direct path by generating fluent policy summaries from raw opinions. Yet prior work highlights two risks in deliberative contexts: (i) as summarizers, LLMs may over-represent majority views while neglecting minority perspectives (Small et al., 2023; Li et al., 2024); and (ii) as evaluators, general-purpose LLMs are inconsistent and biased (Huang et al., 2024; Ye et al., 2024; Thakur et al., 2024; Krumdick et al., 2025), raising concerns about the reliability of current practices. These issues motivate our central question: *Can LLMs truly support deliberation by producing summaries that are representative, informative, and neutral for policy use—and how can we evaluate them reliably at scale?*

To address this gap, we create a framework for automating the evaluation of deliberation summarization at a large scale. As shown in Figure 2, at the first step, we design 10 deliberation questions spanning technology, social media, and public policy, and recruit 300 U.S.-based participants per question to provide free-form opinions (3,000 in total). We then generate summaries from these opinions using multiple LLMs under a multi-scale setting that varies the number of inputs. At the second step, to enable reliable large-scale evaluation, we recruit an additional 4,500 annotators (450 per question); each submits a personal opinion and evaluates a corresponding summary along four deliberation-relevant dimensions: *Representativeness*, *Informativeness*, *Neutrality*, and *Policy Approval*. This process results in 3,000 collected opinions and 4,500 evaluation instances, which together constitute our large-scale human-grounded dataset, DELIBERATIONBANK. Based on this dataset, we systematically evaluate LLM-as-judge approaches and develop DELIBERATIONJUDGE, a fine-tuned DeBERTa-based (He et al., 2020) model for personalized summarization evaluation. Compared to general-purpose LLMs, DELIBERATIONJUDGE achieves closer alignment with human judgments and offers up to 100× greater efficiency.

Using DELIBERATIONJUDGE, we benchmark the summarization performance of 18 LLMs using the 3,000 opinions collected in the first stage. We also conduct a systematic study to examine how factors such as model choice, input configuration, and evaluation method influence the effectiveness of deliberation support through summarization. Our analysis reveals three critical insights. First, off-the-shelf LLM judges exhibit only limited correlation with human judgments across dimensions, with agreement varying by model size. Second, LLM-generated summaries tend to under-represent minority stances, indicating systematic biases that are particularly problematic in deliberative contexts. Third, a lightweight, preference-aligned DeBERTa judge improves reliability and enables large-scale, cost-effective benchmarking of LLM summarizers for deliberation tasks. Our core contributions are as follows:

- We develop DELIBERATIONBANK, a large-scale deliberation benchmark dataset created by 7,500 participants from a US representative sample with two subsets: (i) a public opinion dataset of 3,000 free-form opinions collected from 10 societal deliberation questions on trending topics, and (ii) a summary judgement dataset of 4,500 annotations that evaluate deliberation summaries from individual perspectives.

- We develop an automated evaluation framework to access LLMs' ability to summarize large-scale deliberations and propose DELIBERATIONJUDGE, a fine-tuned DeBERTa model that can judge deliberation summaries from individual perspectives. DELIBERATIONJUDGE outperforms other LLM judge baselines on both alignment with humans and inference efficiency.

- We conduct a rigorous and comprehensive study of LLM summarizers that examines how factors such as opinions input size and deliberation topic shape performance, identifies systematic biases such as order bias and minority stance under-coverage, and clarifies how the two types of minority perspectives differ and why these differences lead to bias.[Dtym]

## 2 DATASET AND EVALUATION SETTING

### 2.1 OVERALL EVALUATION PIPELINE

The overall goal is to evaluate LLM's capabilities to generate representative and effective summaries for public deliberations. We create four deliberation-relevant metrics , grounded in prior work on summarization and societal deliberation (Small et al., 2023; Vijay et al., 2024; Lee et al., 2022). *Representativeness*: Degree to which an individual opinion is semantically captured by the summary, ranging from not represented at all to fully represented; *Informativeness*: How well the summary conveys diverse, detailed, non-redundant information; *Neutrality*: The summary avoids bias and subjective language, maintaining a balanced stance. *Policy Approval*: Perceived suitability of the summary for use by policymakers in decision-making.

To assess the ability of LLMs to generate high-quality deliberation summaries, we adopt an automatic evaluation framework. For each deliberation question, we collect opinions and sample a subset for the summarization model to generate a summary. We evaluate each summary by pairing it with one of the input human opinions and scoring this pair using DELIBERATIONJUDGE, our DeBERTa-based judge fine-tuned on human-annotated data. The judge outputs four continuous scores in $[0, 1]$, corresponding to *representativeness, informativeness, neutrality, and policy approval,* where higher values indicate stronger performance on each criterion.[Dtym]

### 2.2 DELIBERATIONBANK

The public opinion dataset $\mathcal{D}$ is constructed in three stages.

**Stage 1: Collect Initial Opinions.** We begin by constructing a set of ten deliberation questions covering trending societal topics in technology, social media, and public policy; details on question construction are provided in Appendix E.1.[pDNN] These questions fall into two categories: *Open-Ended*, which allow unrestricted free-form responses, and *Binary*, which explicitly frame opinions around two opposing stances (e.g., support vs. oppose). For each question, we recruit a representative pool of 300 U.S.-based participants to provide written opinions. To avoid cross-question contamination and maintain independence across topics, each question is assigned its own participant pool, ensuring that no individual contributes opinions to more than one question.[Dtym]

**Stage 2: Summary Generation for Human Judgment.** We generate summaries using five LLMs spanning different architectures and scales: `GPT-5`, `Claude-4 Opus`, `Gemini-2.5 Pro`, `DeepSeek-V3.1`, and `Qwen3-32B`. For each deliberation question, we apply a multi-scale summarization strategy by providing the model with subsets of the corresponding opinion set of different sizes (e.g., 10, 20, 30, and so on). For each subset size, the model is run three times independently to capture variation in its outputs, producing three summaries per configuration. Across all questions, models, input scales, and resamplings, this stage yields a total of 750 summaries. [Dtym]

**Stage 3: Human Judge Annotation.**

In the final stage, we collect human evaluations for the 750 LLM-generated summaries using a new participant pool that is entirely distinct from Stage 1. The annotation pipeline is implemented using POTATO (Pei et al., 2022). Following prior findings that single-annotator labels can provide reliable supervision in large-scale crowdsourcing settings (Sheng et al., 2008; Snow et al., 2008; MacAvaney & Soldaini, 2023), each annotated instance in our setup is assigned to exactly one annotator.[pDNN]

For each summary, we recruit a dedicated group of six annotators. Each annotator first writes an individual opinion on the underlying question, providing their personal perspective. They then complete two evaluation tasks:[Dtym]

**Rating task.** Annotators rate the summary on four dimensions: *representativeness*, *informativeness*, *neutrality*, and *policy approval*, using a 1–5 scale.

**Comparison task.** Each summary is paired with another summary for the same question (selected using ring-based matching; see Appendix F.1). Annotators compare the two summaries along the same four dimensions, again using a 1–5 scale. [Dtym]

Together, these tasks yield six sets of ratings and comparisons per summary. Aggregated across all 750 summaries, this stage produces a total of 4,500 annotated instances. Detailed guidelines for both annotation tasks are provided in Appendix G.1.[Dtym]

As in all stages involving human participants, data quality checks are applied to filter unreliable responses (e.g., abnormally short completion times; Appendix E.3). These complementary tasks capture both the absolute quality of individual summaries and the relative preference between alternatives, providing rich supervision for judge training.

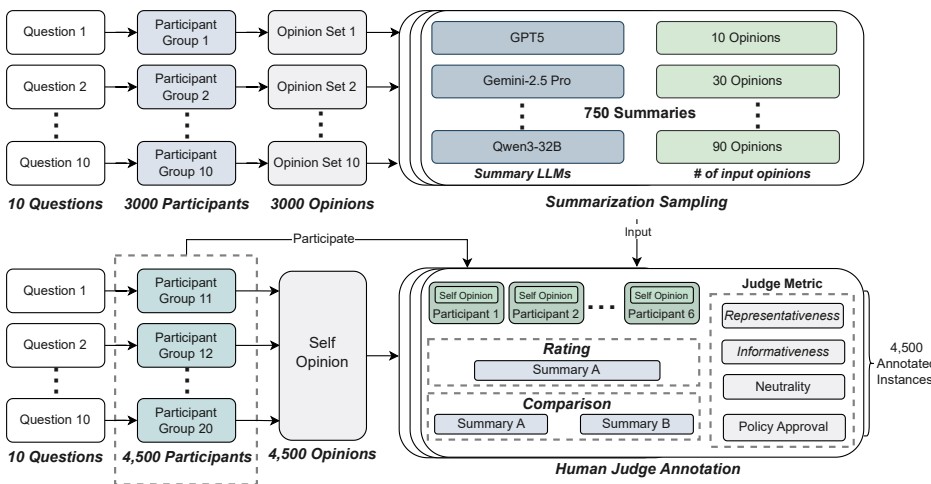

Figure 2: DELIBERATIONBANK creation pipeline

**Final statistics.** Stage 1 yields a benchmark set of 3,000 human-written opinions across the ten deliberation questions. Stage 3 produces 4,500 annotated instances, each accompanied by a new annotator-provided opinion, resulting in 4,500 additional opinions.[Dtym] For downstream use, the annotated instances are randomly split into an 80/20 training–test partition, resulting in 3,600 training instances and 900 test instances. Table 1 summarizes the dataset statistics and provides unified notation for convenient reference in the following sections.[Dtym]

The $\mathcal{T}_{\text{annotate}}$ spans ten predefined deliberation topics, and annotators are uniformly and randomly assigned to exactly one topic. Each topic is completed by a distinct group of 450 annotators, resulting in non overlapping annotator pools across topics and ensuring that instances for different topics originate from independent contributor groups.[wcNP]

Table 1: Data statistics across three stages.

|  | Question List | Benchmark Opinions | Annotated Instances | Annotator Opinions | Train Split (80%) | Test Split (20%) |
|---|---|---|---|---|---|---|
| Notation | $\mathcal{Q}$ | $\mathcal{D}_{\text{bench}}$ | $\mathcal{T}_{\text{annotate}}$ | $\mathcal{D}_{\text{annotate}}$ | $\mathcal{T}_{\text{train}}, \mathcal{D}_{\text{train}}$ | $\mathcal{T}_{\text{test}}, \mathcal{D}_{\text{test}}$ |
| Count | 10 | 3,000 | 4,500 | 4,500 | 3,600 | 900 |

## 3 AUTOMATING JUDGEMENT OF LLM SUMMARIES WITH DELIBERATIONJUDGE

Previous works in public consensus generation often incorporate an auxiliary judging mechanism to evaluate model outputs. For example, Generative Social Choice (Fish et al., 2023) leverages discriminative LLM queries together with social-choice aggregation to produce representative proposal slates, whereas Bakker et al. (2022) trains a reward model that predicts individual approval and guides supervised fine-tuning toward high-consensus statements. Building on these prior works, we argue that a judge model is necessary for automated evaluation in our LLM-based deliberation summarization benchmark task. Such a judge should satisfy two requirements:

**Reliability.** aligned with the principle demonstrated in Bakker et al. (2022), the judge should be aligned with human preference and reflect representational fidelity rather than unconstrained behavior from a vanilla LLM. Moreover, empirical studies show that LLM-as-judge paradigms often exhibit instability, preference skew, and demographic or ideological bias (Huang et al., 2024; Ye et al., 2024; Thakur et al., 2024; Krumdick et al., 2025; Yang et al., 2025), reinforcing the need for a reliable, preference-grounded judge.

**Efficiency.** As a benchmarking evaluator must operate repeatedly at the instance-level scale, the judge must remain lightweight enough to support large-volume scoring without incurring prohibitive computational cost. [Dtym]

To balance these two features, we introduce DELIBERATIONJUDGE, a DeBERTa-based model fine-tuned on a large-scale human judgment dataset tailored to deliberation summarization, which we subsequently employ for automatic evaluation.

### 3.1 LIMITED CORRELATIONS BETWEEN HUMAN AND LLM JUDGMENTS

To examine the consistency between human and LLM judgments, we take the same inputs used in the human-annotated test set, each consisting of a question, an annotator-written opinion, a target summary, and a comparison summary, then we ask LLMs to provide annotations following the identical guidelines given to human annotators. The LLMs do not generate new opinions; they only re-annotate the existing items. This process produces an LLM-labeled test set that is directly comparable to the human-labeled version, enabling a straightforward correlation analysis between human and model judgments.[Dtym]

After annotation, we compute correlations between LLM and human judgments. As shown in Figure 3, larger models achieve higher agreement with human judges, while very small models (e.g., Qwen3-0.6B, Qwen3-1.7B) perform poorly across all dimensions. In the rating task, alignment is strongest on *Representativeness* and *Policy Approval*, with correlations around 0.30–0.35, suggesting these dimensions are easier for LLMs to approximate. In the comparison task, overall correlations are lower, though *Informativeness* reaches about 0.37. Overall, correlations never exceed 0.4, showing that while LLMs can partially approximate human judgments, off-the-shelf models remain insufficient as automated judges and motivate the need for more reliable, dedicated judge models.

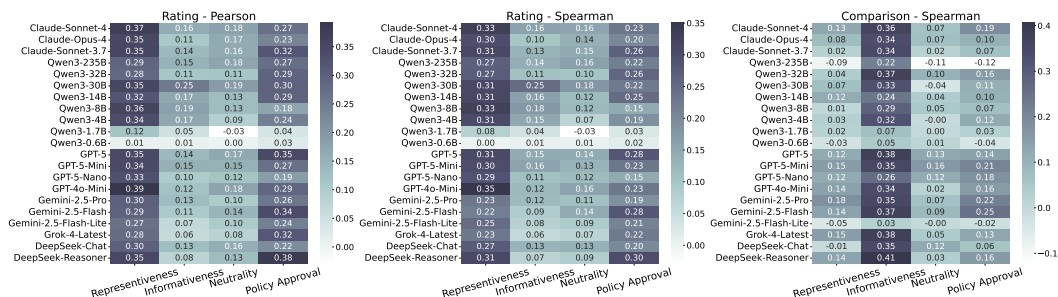

Figure 3: Heatmap of consistency between human and LLM judges on two judgment tasks. **Left**: Rating–Pearson $r$; Center: Rating–Spearman $\rho$; Right: Comparison–Spearman $\rho$. Lighter to darker colors represent lower to higher correlations.

### 3.2 DELIBERATIONJUDGE

Considering the lack of consistency between LLM as Judge and human annotation, we propose DELIBERATIONJUDGE, a DeBERTa-based judge $\mathcal{J}_\theta$, trained with supervised fine-tuning (SFT) on the two types of human annotations contained in the training set $\mathcal{T}_{\text{train}}$ [Dtym] introduced in § 2.2. To place both rating and comparison tasks on a unified scale, we normalize the labels for each evaluation dimension. Specifically, five-point Likert scores from the rating task (Summary A alone) are retained on $[1, 5]$, while five-point relative judgments from the comparison task (Summary A vs. Summary B) are linearly mapped into the extended interval $[-1, 7]$. This normalization allows both annotation sources to be trained jointly as graded supervision on the same scale. Formally, given an input consisting of a deliberation question $q_i$, an annotator opinion $o_S^{(a)}$, and a candidate summary $S_{\mathcal{M}, \tilde{\mathcal{O}}_i}$ produced by model $\mathcal{M}$ from opinion subset $\tilde{\mathcal{O}}_i$, the judge $\mathcal{J}_\theta$ encodes the concatenated sequence

$$[\text{CLS}] \; q_i \; [\text{SEP}] \; o_S^{(a)} \; [\text{SEP}] \; S_{\mathcal{M}, \tilde{\mathcal{O}}_i} \; [\text{SEP}]$$

and outputs a four-dimensional normalized score vector:

$$\hat{\mathbf{y}} = \mathcal{J}_\theta\big(q_i, o_S^{(a)}, S_{\mathcal{M}, \tilde{\mathcal{O}}_i}\big) = \big(\hat{y}^{(\text{rep})}, \hat{y}^{(\text{inf})}, \hat{y}^{(\text{neu})}, \hat{y}^{(\text{pol})}\big) \in [0, 1]^4.$$

Here the [CLS] representation from the final encoder layer is passed through a hidden layer and a linear projection to produce the four regression outputs. Human annotations $\mathbf{y}^{\text{raw}} \in [-1, 7]^4$ are

linearly normalized to $\mathbf{y} \in [0,1]^4$ for training stability. The model is trained with the Huber loss averaged across dimensions:

$$\mathcal{L}(\theta) = \frac{1}{|\mathcal{T}_{\text{train}}|} \sum_{(q,o,S) \in \mathcal{T}_{\text{train}}} \frac{1}{4} \sum_{d \in \{\text{rep,inf,neu,pol}\}} \ell_\delta\left(\hat{y}^{(d)}, y^{(d)}\right),$$

where

$$\ell_\delta(\hat{y}, y) = \begin{cases} \frac{1}{2}(\hat{y} - y)^2 & \text{if } |\hat{y} - y| \leq \delta, \\ \delta \cdot \left(|\hat{y} - y| - \frac{1}{2}\delta\right) & \text{otherwise.} \end{cases}$$

At inference time, predictions remain in the $[0,1]$ range and are used directly as summary scores. Detailed training settings are provided in Appendix H.1. We also evaluated several backbones and found `DeBERTa-v3-base` to perform best. Full comparison results are provided in Appendix H.2.[GiWi]

### 3.3 DELIBERATIONJUDGE OUTPERFORMS SOTA LLMs IN CONSISTENCY AND EFFICIENCY

To verify that DELIBERATIONJUDGE effectively improves both reliablity and efficiency, we evaluate the agreement between human judgments and DELIBERATIONJUDGE, and compare it against the Human–LLM correlations reported in §3.1. For this analysis, we use the same held-out test set that was previously used for LLM-based annotation. Each item in this test set also includes a question, an annotator-written opinion, and two summaries generated for the same question. Because DELIBERATIONJUDGE evaluates one summary at a time, every test item is split into two separate regression examples, one for each of the two summaries, while keeping the outputs normalized to $[0,1]$. With 900 original test items, this conversion produces 1,800 instances each with a question, an opinion and a summary, forming the new derived test set $\hat{\mathcal{T}}_{\text{test}}^{\text{Delib}}$ used in our correlation analysis. [Dtym]

We then examine the average Spearman correlation of automatic judges (LLMs and DELIBERATION-JUDGE) with human annotations[pDNN], as well as their efficiency in producing judgments for each (question, opinion, summary) pair. In Figure 4 (left), we observe that DELIBERATIONJUDGE achieves the highest scores in both agreement with human preferences and efficiency. Its overall correlation reaches approximately 0.48, outperforming the second-best model, `Claude-3.7-Sonnet`, which achieves only around 0.20. In terms of efficiency, DELIBERATIONJUDGE requires only about 0.03 seconds per item, while most LLM judges take more than 8 seconds on average. Although `Qwen3-0.6B` approaches our method in inference speed, it performs poorly in judgment quality. From another

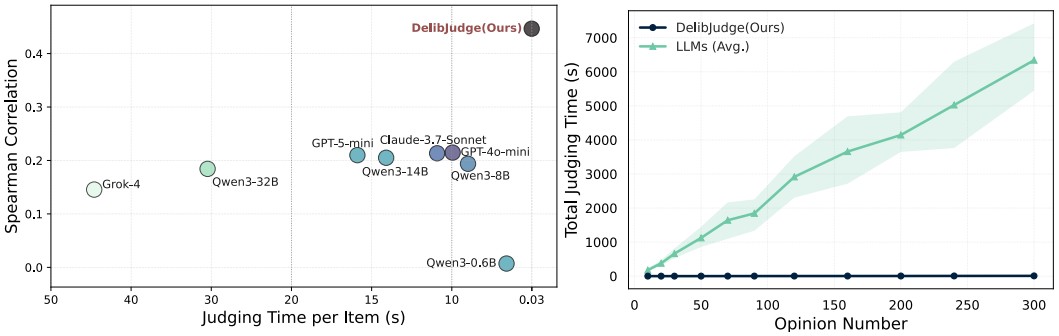

Figure 4: **Left**: Judging Time vs. Spearman Correlation. DELIBERATIONJUDGE achieves both the lowest time and the highest correlation compared to LLMs.[pDNN] **Right**: Scaling Stability. Total judging time of DELIBERATIONJUDGE and eight LLMs as the number of comments increases; solid lines show means and shaded areas denote min–max ranges. In the figures, DelibJudge=DELIBERATIONJUDGE.

perspective, efficiency becomes even more critical when scaling to large deliberation settings. For example, if we need to evaluate summaries conditioned on thousands of deliberation opinions, the inference time cost quickly becomes prohibitive. As shown in Figure 4 (right), we compute the total time required by DELIBERATIONJUDGE and the average of eight LLM judges as the number of input opinions increases. Since judgments must be executed once for each (question, opinion, summary) pair, the total cost for LLMs grows rapidly with opinion number. In contrast, the total time for DELIBERATIONJUDGE remains largely unaffected by input size, owing to its fast per-pair processing

speed. We additionally report DELIBERATIONJUDGE's performance on OOD data in Appendix H.3, which illustrates how its behavior extends beyond the training domains and in Appendix A we discuss possible directions for further usage and strengthening of DELIBERATIONJUDGE.[wcNP]

### 3.4 STRONG RANK CORRELATION BETWEEN DELIBERATIONJUDGE AND HUMAN JUDGMENTS

Table 2: Spearman rank correlation (human vs. DELIBERATIONJUDGE) across four dimensions.

|              | Average | Representativeness | Informativeness | Neutrality | Policy Approval |
|--------------|---------|--------------------|-----------------|------------|-----------------|
| Spearman $\rho$ | 0.70    | 0.60               | 1.00            | 0.90       | 0.90            |

We evaluate how well DELIBERATIONJUDGE tracks human judgments on the five models used in Stage 2 summarization for building DELIBERATIONBANK (§2.2). Concretely, for each evaluation dimension and overall average score, we compute the Spearman rank correlation across models between human scores and DELIBERATIONJUDGE predictions on $\hat{\mathcal{T}}_{\text{test}}^{\text{Delib}}$ and the original $\hat{\mathcal{T}}_{\text{test}}$. As shown in Table 2, DELIBERATIONJUDGE exhibits strong rank alignment overall ($\rho = 0.70$), with near-perfect agreement on Informativeness ($\rho = 1.00$) and high alignment on Neutrality and Policy Approval ($\rho = 0.90$ each). Representativeness shows moderate alignment ($\rho = 0.60$), reflecting minor ordering differences in that dimension.

## 4 BENCHMARKING LLMS FOR DELIBERATION SUMMARIZATION

We evaluate a broad set of language models, including both proprietary frontier systems and open-weight models spanning a wide range of parameter scales (see Table 6 in Appendix D). For each deliberation question, we use the full set of 300 opinions collected in Stage 1 and construct multiple input conditions by sampling subsets of sizes 10, 20, 30, 50, 70, 90, 120, 160, 200, 240, and 300. Given a sampled subset, each model generates three summaries through independent runs to capture sampling variability. Each summary is then evaluated by DELIBERATIONJUDGE against every opinion in the same subset, producing four scores, each normalized to the range $[0, 1]$. Finally, the results from the three independently generated summaries are aggregated, and 95% confidence intervals are computed for each evaluation metric. [Dtym]

Table 3: Overall average performance, (mean± 95% CI half-width) across four evaluation dimensions (Representativeness, Informativeness, Neutrality, and Policy Approval). Models are arranged from left to right and top to bottom in descending order of mean performance.

| Model          | Overall           | Model        | Overall           | Model            | Overall           |
|----------------|-------------------|--------------|-------------------|------------------|-------------------|
| GPT-5-Mini     | $0.622 \pm 0.004$ | GPT-5        | $0.617 \pm 0.005$ | Claude-Sonnet-4  | $0.615 \pm 0.004$ |
| Claude-Opus-4  | $0.614 \pm 0.004$ | Qwen3-32B    | $0.609 \pm 0.006$ | Gemini-2.5-Flash | $0.605 \pm 0.004$ |
| Grok-4         | $0.594 \pm 0.003$ | Gemini-2.5-Pro | $0.594 \pm 0.005$ | Qwen3-235B     | $0.593 \pm 0.004$ |
| Qwen3-14B      | $0.584 \pm 0.003$ | Qwen3-1.7B   | $0.583 \pm 0.005$ | DeepSeek-Reasoner | $0.580 \pm 0.004$ |
| DeepSeek-Chat  | $0.576 \pm 0.003$ | Qwen3-4B     | $0.575 \pm 0.004$ | Qwen3-8B         | $0.575 \pm 0.003$ |
| GPT-4o-Mini    | $0.571 \pm 0.003$ | Qwen3-30B    | $0.571 \pm 0.003$ | Qwen3-0.6B       | $0.550 \pm 0.004$ |

### 4.1 BENCHMARKING RESULTS[wcNP]

As shown in Table 3, the `GPT-5` and `Claude-4` families lead the leaderboard, averaging 0.61–0.62 across four dimensions. The `Qwen3-32B` outperforms several closed-source models (e.g., `Gemini-2.5`, `Grok-4`), mainly due to its strong *Neutrality* (Figure 5, lower left). Smaller models such as `Qwen3-0.6B` perform worst, consistent with scaling trends. Overall differences are modest: the gap between best and worst is $< 0.1$, indicating that most models capture deliberation summaries at a broadly similar quality level. Figure 5 shows similar per-dimension trends, with *Neutrality* tightly clustered ($\sim 0.02$ spread) and the other three dimensions showing wider gaps (0.05–0.1).

### 4.2 ANALYSIS OF FACTORS AFFECTING MODEL PERFORMANCE

**Summarization Input Size.** Each deliberation question has 300 opinions, which we partition into subsets of varying sizes to test input effects. As shown in Figure 6 (upper left), the model's average score at each opinion subset size $n$ improves consistently as $n$ increases from 10 to about 100.

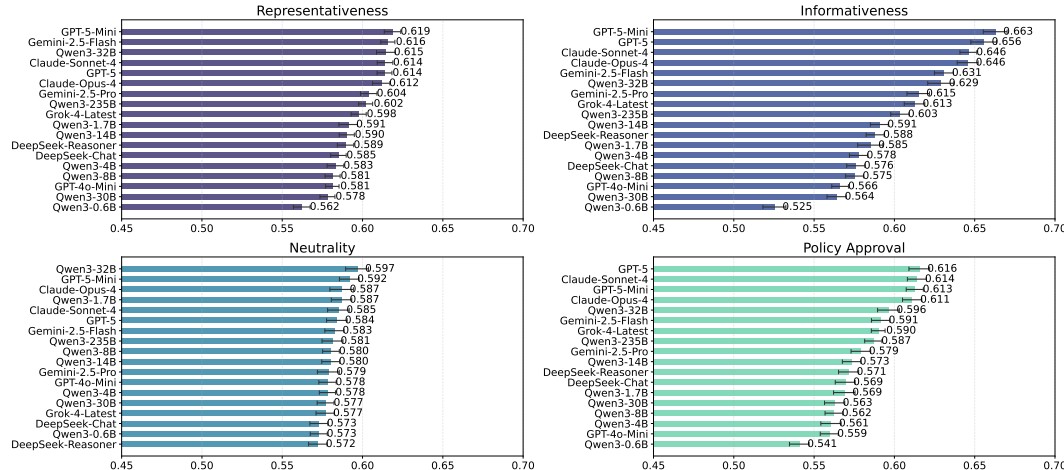

Figure 5: Comparative performance on Representativeness, Informativeness, Neutrality, and Policy Approval (mean ± 95% CI); higher is better.

beyond which it plateaus. Thus, while LLMs benefit from moderate input sizes, they fail to leverage substantially larger sets, revealing limited scalability in handling large deliberation contexts and underscoring the need for more effective aggregation mechanisms. We further examine whether the ordering of opinions affects model performance. As shown in Appendix B, we do not observe any meaningful order sensitivity from model summarizations.[Dtym]

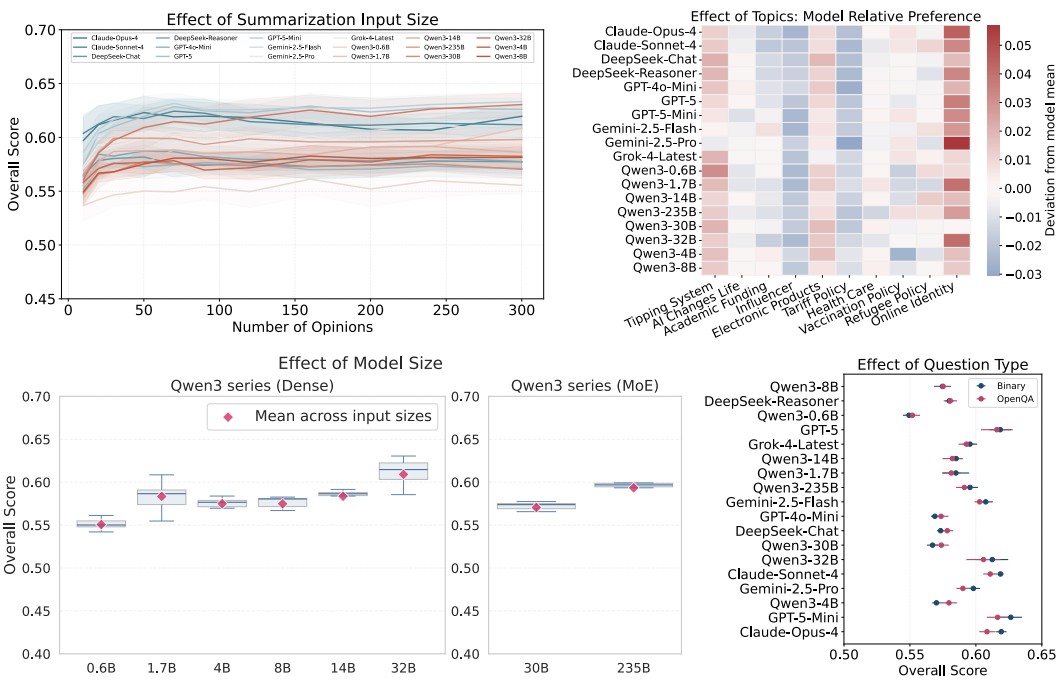

Figure 6: Overview of factors affecting model performance. **Upper Left**: Effect of input subset size with $n \in \{10, 20, \ldots, 300\}$. **Upper Right**: Topic heatmap across models, where each cell shows the topic-wise average score after centering by subtracting the model's global mean (red = above global mean; blue = below). **Lower Left**: Effect of model size, showing overall score of the `Qwen3` family, comparing Dense and MoE architectures from 0.6B to 235B. **Lower Right**: Effect of question type (Binary vs. Open-Ended), with models ranked by $|\Delta|$, the absolute difference between type-wise average scores (top = most robust).

**Topics.** We further analyze model sensitivity across topics by computing relative preference scores, defined as, over the full dataset, the difference between the question-level average score (for question $q$) and the model's global average score (pooled across all questions, subset sizes, and runs); (see Appendix G.2 for the formal definition). As shown in Figure 6 (upper right), topics such as *Online Identity* and *Tipping System* yield higher-than-average scores due to more concentrated opinions, whereas open-ended topics like *Influencers* produce dispersed views, making it harder for models to capture all perspectives. These results suggest that sensitivity depends not only on question type but also on framing, with narrower questions yielding more consistent summaries.

**Question Type.** Our benchmark includes ten deliberation questions, evenly split between binary and open-ended types. Figure 6 (lower right) presents a type-wise comparison of model average performance, ranking systems by $|\Delta|$, the absolute difference between average scores on Binary and Open-Ended questions; smaller $|\Delta|$ indicates lower sensitivity. Most smaller models (except Qwen3-4B) are consistent across types, though at low absolute scores. GPT-5 combines high average performance with relatively small sensitivity. Overall, 11 of 18 models perform better on binary questions, likely because binary framing constrains responses to two clear positions, while open-ended prompts elicit diverse opinions that are harder to summarize.

**Model Size.** We next examine the effect of model size on summarization performance. Figure 6 (lower left) shows boxplots of each Qwen3 model's overall score comparing Dense (Xiao et al., 2024) and Mixture-of-Experts (MoE) (Mu & Lin, 2025) architectures. In line with scaling law observations in §4.1, larger models achieve higher scores: within the dense series, performance rises steadily from 0.6B to 32B, while in the MoE series, the 235B model surpasses its 30B counterpart. Gains from scaling are evident but not perfectly monotonic, as smaller models (e.g., 1.7B) show fluctuations, suggesting that architectural design and training stability also influence performance beyond raw parameter count.

## 5 CASE STUDY: HOW WELL CAN LLMs REPRESENT MINORITY OPINIONS?

To examine whether LLMs face challenges in representing minority opinions, we conducted a focused case study on two representative deliberation questions: a Binary question on *Tariff Policy* and an Open-Ended question on *AI Change Life*.

### 5.1 SUBJECTIVELY DEFINED MINORITY

**Self-Reported Minority Data Collection.** For each question, we collected 1,000 opinions from a new pool of U.S. participants. In addition to providing their stance on the question, participants were explicitly asked to self-identify whether they believed their opinion belonged to a minority group (i.e., *Do you think your opinion differs from that of most people in the U.S.?*). Three response options were provided: *Yes*(Positive), *No*(Negative), and *I'm not sure*(Netural). This self-reported annotation enables a relative ground-truth partition of the data into minority and non-minority subsets. Compared to earlier collection settings with 300 responses, we increased the sample size to 1,000 to ensure more reliable minority/non-minority estimates.

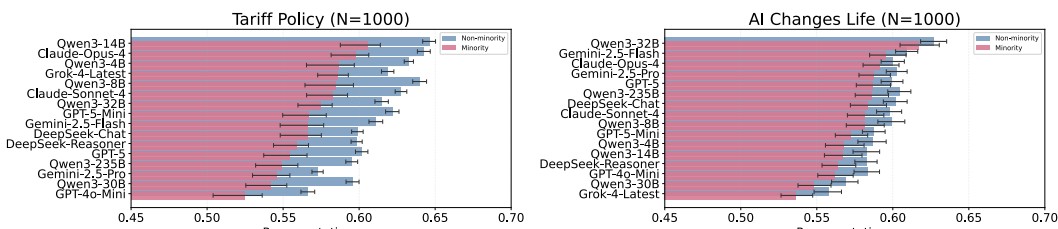

Figure 7: Comparison of model representativeness scores for minority and non-minority opinions using *annotator self-identification* on two representative deliberation questions. Results are reported using a summarization input size of 1000.

**Key Findings.** We focus on the *representativeness* dimension, as it best reflects whether an individual opinion is covered by the generated summary. Based on participant self-reports, we partitioned the dataset into minority and non-minority subsets, treating only responses marked as *Yes* as minority and all others as non-minority, and then computed the average representativeness score for each

model on the two groups using 1,000 opinions as summarization input. As shown in Figure 7, across all models, representativeness scores for non-minority opinions are higher than those for minority opinions, revealing a systematic bias. The effect is strongest in the binary *Tariff Policy* case (gap up to 0.08) and smaller in the open-ended question (about 0.02). We attribute this disparity to the relative scarcity of minority responses in the *Tariff Policy* setting (see the minority distribution in Figure 10) , which makes models more likely to overlook them.

## 5.2 OBJECTIVELY DEFINED MINORITY

**Automatic Outlier Detection.** To strengthen the reliability of our analysis, we introduce an alternative minority identification method based on semantic outlier detection. Instead of relying solely on self reported perceptions, we identify opinions that occupy sparse regions of the embedding space. All responses are encoded using `all-MiniLM-L6-v2`, and outliers are selected via the Local Outlier Factor (LOF) with 20 neighbors. For the binary question *Tariff Policy*, where stance labels are available, we first group responses into positive, negative, and neutral categories, then apply LOF within each group and mark the top ten percent as semantic minorities. This surfaces opinions that share the same stance but use uncommon reasoning or framing. For the open ended question *AI Changes Life*, which lacks stance structure, we cluster embeddings into eight coarse themes using k-means and apply LOF within each cluster, again selecting the top ten percent as outliers. This captures opinions that deviate from typical expressions within each theme. Additional procedural details are included in Appendix C.2.

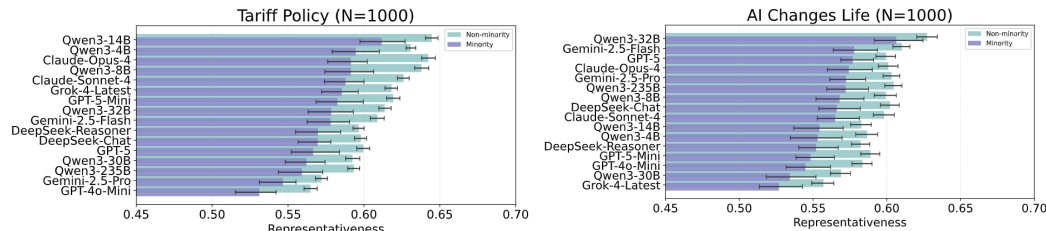

Figure 8: Comparison of model representativeness performance for minority and non-minority opinions across two representative deliberation questions using *automatic outlier detection*. Results are reported with a summarization input size of 1000.

**Key Findings.** Similar to prior self-reported setting, we focus on *Representiveness* and using input size of 1000. From Figure 8, our outlier-based analysis reveals patterns that closely mirror those in the subjectively defined minority setting. For most models, automatically identified minority opinions receive consistently lower representativeness scores than majority ones. On AI Changes Life, the average gap is 0.04, and on Tariff Policy it is 0.03. As a sanity check, we also compare against randomly sampled opinion sets with the same minority to non-minority ratio (Figure 12 in Appendix C.3). The resulting representativeness difference is only 0.001 to 0.002, which is over 30x smaller than the gap observed for actual minority opinions.[Dtym]Futhermore, we utilize the automatic minority detection method to analyze the minority bias pattern over all 10 topic, provided in Appendix C.4.[pDNN]

## 6 CONCLUSION

Summarizing public opinions is a key step to enabling large-scale deliberations. While existing studies have demonstrated the potential of LLMs for deliberation summarization, it remains unclear which LLM and what setting lead to the optimal summarization experience. One of the key bottlenecks of deliberation summarization evaluation is the scalability of human evaluation, as the perception of LLM summarization is highly subjective. In this paper, we present DELIBERATIONBANK, a large-scale deliberation and summarization evaluation dataset, and DELIBERATIONJUDGE, a fine-tuned model that can accurately and efficiently scale the judgment of deliberation summarizations. Leveraging the dataset and model, we provide a systematic evaluation of 18 LLMs and our study reveals key insights and limitations of LLM for deliberation summarization.

ETHICS STATEMENT

This work relies on human annotation to evaluate deliberation summaries. All annotators were recruited through the Prolific platform and managed with the Potato annotation system, which ensured complete anonymization of responses and secure task assignment. Participants provided informed consent prior to annotation, and no personally identifiable information (PII) was collected or stored at any stage. Sensitive questions were included solely for research purposes and were framed in a neutral manner. Annotators were compensated at fair market rates.

REPRODUCIBILITY STATEMENT

We provide full details of training hyperparameters, model settings, and evaluation protocols in Appendix H. Upon publication, we will release the detailed dataset information, annotation guidelines, preprocessing scripts, training code, and evaluation scripts to enable independent replication and further study. Together, these resources will allow researchers to reproduce our experiments and extend our framework.

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

# A  FUTURE WORK AND BROADER IMPACT

Our study shows that DELIBERATIONJUDGE better aligns with human evaluators than general purpose LLM judges, though correlation levels remain moderate. Expanding DELIBERATIONBANK beyond the current ten questions would broaden generalizability. In addition, diversity aware or stance conditioned evaluation is a natural extension of our framework. Incorporating objectives that explicitly model viewpoint diversity, such as contrastive formulations across stances or subgroups, may further improve an evaluator's ability to capture fine grained differences in perspective coverage and provide a more nuanced view of deliberative diversity.[wcNP]

We also finds that models systematically assign higher representativeness scores to majority viewpoints than to self identified minority opinions, suggesting that fairness constraints may improve minority coverage. Also, the human written opinions in DELIBERATIONBANK provide a large and structurally unified corpus for studying alignment, bias, minority representation, robustness, diversity aware summarization, and judgment or reward modeling. The DELIBERATIONJUDGE can likewise serve as a deliberation specific evaluator or be fine tuned as a reward model to support socially aware, representativeness preserving summarization systems.[pDNN]

# B  ORDER BIAS ANALYSIS OF LLMS[Dtym]

By analyzing the contributions from different positions, we infer whether the model is sensitive to input order bias. We inspect positional effects using 2,000 newly collected annotations, with 1,000 instances for each of the two topics described in §5. For each prompt, we construct two input prefixes containing the first 500 and the full 1,000 comments, which produces four evaluation splits for each summarization model. Within each split, we compute the average representativeness score at each list position across resampled summaries, convert indices to 1-based positions, and smooth the trajectory by grouping every twenty positions into bins and plotting the mean with standard error. We repeat this procedure for two summarizers, GPT-5-mini and Qwen3-30B-a3b. Across both models, the curves do not display any consistent upward or downward pattern. Instead, the trajectories fluctuate without a clear monotonic direction, suggesting that positional effects are not a major source of bias in our setting.[Dtym]

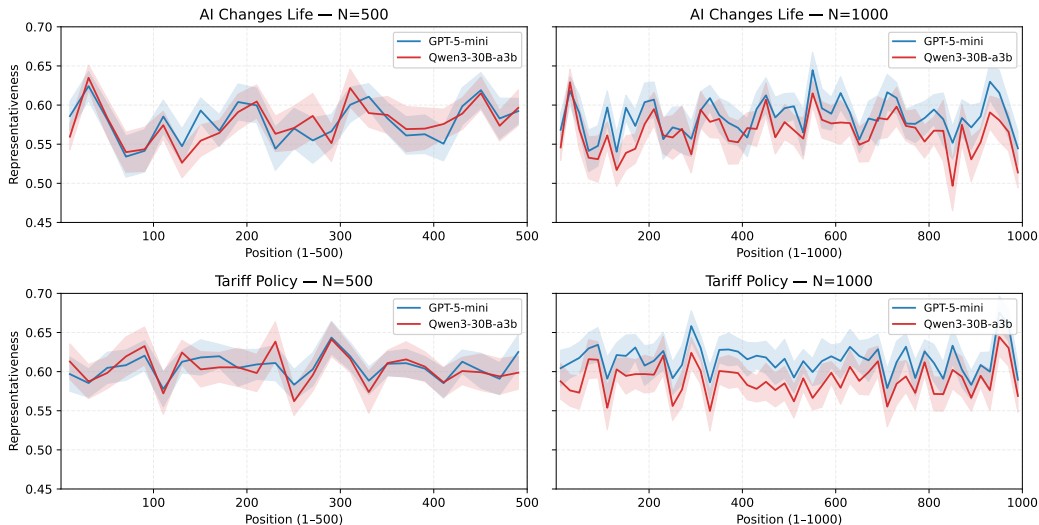

Figure 9: Representativeness trajectories (mean ± SE over 20-position bins) for GPT-5-mini and Qwen3-30B-a3b when summarizing the first 500 and 1,000 comments of each topic.[Dtym]

## C  FURTHER EXPERIMENTAL DETAILS AND RESULTS MINORITY BIAS

### C.1  SUBJECTIVELY DEFINED MINORITY

AI Changes Life

| Yes 19% | No 47% | I'm not sure 33% |

Tariff Policy

| Yes 11% | No 60% | I'm not sure 28% |

Figure 10: Distribution of minority opinions in the newly data for two representative questions.

### C.2  OBJECTIVELY DEFINED MINORITY

**Preliminary 1: Local Outlier Factor (LOF).**

In this preliminary, we include the formal definition of the Local Outlier Factor (LOF) used in our semantic minority detection pipeline. For each opinion embedding $x$, let $N_k(x)$ denote its $k$-nearest neighbors. The *local reachability density* (LRD) of $x$ is defined as:

$$\text{LRD}_k(x) = \left( \frac{1}{|N_k(x)|} \sum_{y \in N_k(x)} \max\{d(x,y), \text{k-dist}(y)\} \right)^{-1},$$

where $d(\cdot, \cdot)$ is the Euclidean distance and k-dist$(y)$ is the distance from $y$ to its $k$-th nearest neighbor.

The *local outlier factor* of $x$ is then:

$$\text{LOF}_k(x) = \frac{1}{|N_k(x)|} \sum_{y \in N_k(x)} \frac{\text{LRD}_k(y)}{\text{LRD}_k(x)}.$$

Values $\text{LOF}_k(x) > 1$ indicate that $x$ lies in a region of lower density relative to its neighbors and is therefore more likely to be an outlier.

**Preliminary 2: Representativeness Bias.** We define the representativeness bias of opinion set $A$ relative to set $B$ as the difference between their average predicted representativeness. Let $\hat{y}^{(\text{rep})}(o)$ denote the representativeness score assigned to opinion $o$. The bias is

$$\Delta_{\text{rep}}(A, B) = \frac{1}{|B|} \sum_{o \in B} \hat{y}^{(\text{rep})}(o) - \frac{1}{|A|} \sum_{o \in A} \hat{y}^{(\text{rep})}(o).$$

Positive values indicate that set $A$ receives lower representativeness than set $B$. In our analysis, we focus on two instantiated forms of the above metric. The *minority bias* is defined as

$$\Delta_{\text{rep}}(\text{minority}, \text{non-minority}),$$

which measures the average representativeness gap between minority and non-minority opinion sets.

Similarly, the *random bias* is defined as

$$\Delta_{\text{rep}}(\text{selected}, \text{non-selected}),$$

where the selected group is obtained by random sampling opinions from the full pool at the same proportion as the minority set, and the non-selected group is drawn by sampling the complement at the same ratio. This construction provides a random baseline.[pDNN]

**Automatic Minority Detection.** We introduce an embedding based outlier detection method to identify semantically distinctive minority opinions. All responses are first encoded into 384 dimensional embeddings using the `all-MiniLM-L6-v2` sentence transformer, followed by a grouping step that adapts to the question type. Local Outlier Factor (LOF with 20 neighbors) is then applied within each group to detect semantically rare opinions. This shared embedding and LOF configuration ensures comparability across settings.

For binary questions (i.e. Tariff Policy), we leverage the explicit stance labels collected in the survey. Responses are partitioned into Positive, Negative, and Neutral groups, after which LOF is applied

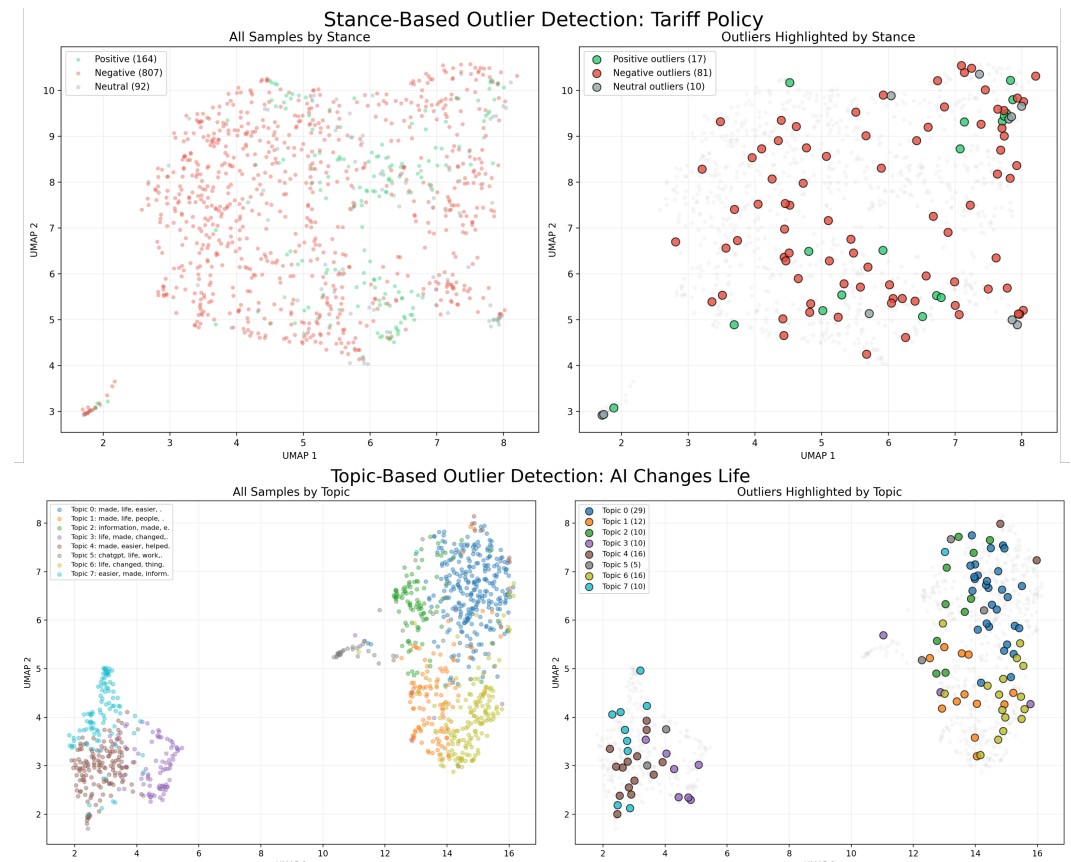

Figure 11: **Top:** Stance based outlier detection for the Tariff Policy question. The left panel shows all samples colored by stance, and the right panel highlights detected stance outliers within the same UMAP space; **Bottom:** Topic based outlier detection for the AI Changes Life question. The left panel shows all samples colored by discovered semantic topics with K-means, while the right panel highlights detected topic outliers.

separately within each stance group. This setup allows us to capture respondents who share the same overall stance but reach it through unusual reasoning styles, such as invoking personal narratives or ideological arguments distinct from the dominant economic framing. We select the top ten percent of responses by LOF score in each stance group, identifying 108 outliers.

For open ended questions (i.e. AI Changes Life), where no stance labels exist, we cluster standardized embeddings into eight topical groups using K-Means. LOF is then applied within each topic cluster to identify responses that convey rare perspectives or exceptional experiences within the same theme. This two stage design prevents mechanically treating surface form variation as outliers and instead focuses on semantic distinctiveness.[Dtym]

### C.3 ZOOM IN: ANALYSIS OF THE TWO TYPES OF MINORITY.

In the previous §5, we observe that minority groups identified through both subjective self-reports and automatic outlier detection exhibit consistently lower overall representativeness than non-minority opinions across summarization models (Figure 7 and Figure 8). To ensure that this effect is not driven by differing group sizes, we conduct a control experiment where opinions are randomly sampled according to the same minority to non minority ratio produced by the automatic detection method. The randomly formed sets show nearly identical average representativeness (Figure 12), confirming that the observed gaps are not artifacts of sampling imbalance.

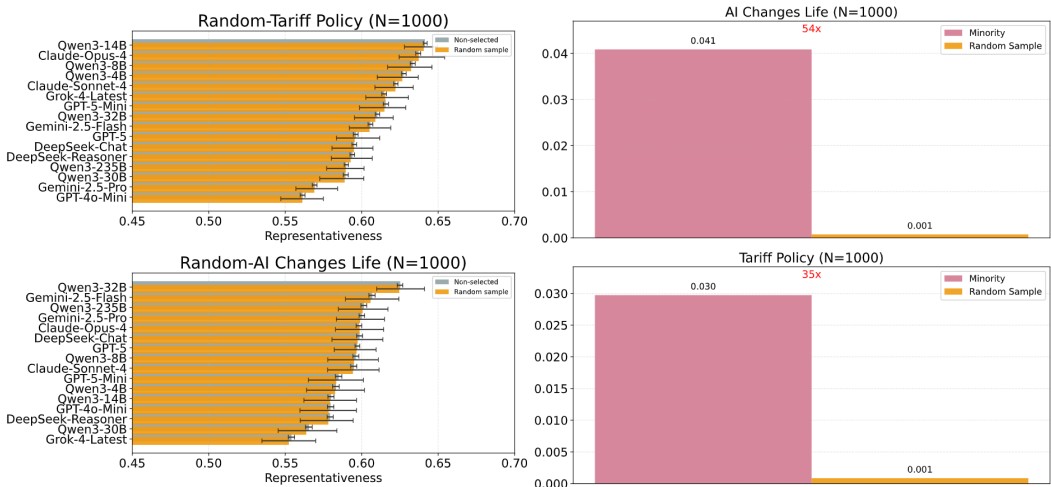

Figure 12: **Left:** Representiveness score of randomly selected opinions and non-selected opinions across all models. Random samples show near-zero bias. **Right:** Comparison of automatically detected minority bias and random bias. Detected minority show 35–54x larger bias.

Table 4: Overlap between subjective (self-reported) and objective (automatic detection) minorities across two topics. Low Jaccard similarity and intersection rate indicate that the two methods capture different populations, supporting that they measure distinct constructs: subjective social perception versus objective semantic uniqueness.

|  | Jaccard Similarity | Intersection Rate |
|---|---|---|
| Tariff Policy | 0.090 | 0.018 |
| AI Changes Life | 0.096 | 0.026 |

Despite these consistent under representation patterns, the two minority sets overlap surprisingly little. As shown in Table 4, both the Jaccard similarity and intersection rate remain below ten percent. This strikingly low overlap raises an important question: *what underlying factors lead to the existence of two distinct minority sets, and in what ways do these sets fundamentally differ?*

To understand the differences between the two minority definitions, we conduct a case study and present representative examples in Table 5. The examples illustrate that self-reported and automatically detected minorities arise from distinct underlying mechanisms. Self-reported minorities reflect a subjective perception of being under represented. These respondents often experience a personal sense of isolation, even when their statements are not semantically unique. Their responses frequently contain high uncertainty language, such as "I'm not sure," reflecting a feeling of difference that emerges entirely from subjective judgment without knowledge of how others answered.

In contrast, automatically detected minorities are defined by objective semantic distinctiveness in the embedding space. These responses contain uncommon argumentation patterns or rare thematic associations, such as linking tariff policy to racial discourse or describing AI as a retirement advisor. Although these perspectives are unusual within the population, respondents themselves typically do not recognize their uniqueness. These perceptual versus structural foundations explain why the two sets overlap by less than ten percent.

Despite their differences, both types of minority opinions exhibit lower representativeness in model generated summaries. Self-reported minorities tend to be expressed with less confidence or clarity, making them more susceptible to being overlooked during summarization. Automatically detected minorities, on the other hand, are semantically rare within their stance or topic groups, causing them to be deprioritized when models focus on dominant patterns. Consequently, both subjective and semantic minorities are systematically under represented, though for different reasons.[wcNP]

Table 5: Representative examples illustrating the conceptual differences between self-reported minorities and automatically detected semantic minorities.

| Minority Type | Representative Example (Tariff Policy / AI Changes Life) | Why This Case? What It Reveals |
|---|---|---|
| Self-Reported | *"I'm not sure if the tariff policy will have a positive or negative impact... more data is needed."* | Strong uncertainty and self-perceived isolation; Semantically similar to many mainstream responses; Minority defined by subjective perception rather than semantic uniqueness |
| | *"AI gives me a judgement-free space to talk about PTSD... I feel alone and unsupported."* | Personal narrative and emotional framing; Overlaps with common "AI for support" themes; Shows how individuals feel minority due to personal context |
| Automatically Detected | *"Tariffs help restart conversations about race and ethnicity... this is good for public discourse."* | Rare conceptual framing (tariffs → racial discourse); Distinct reasoning but author does not feel minority; Minority defined by semantic outlier status |
| | *"AI helps me grow my own SCOBY and ferment kombucha; I use it for very specific tasks."* | Highly specific and unusual use case; Not self-identified as minority; Shows how semantic uniqueness does not imply self-awareness |

### C.4 MINORITY BIAS PATTERNS ACROSS ALL TOPICS

Using the previously introduced automatic minority detection, we extend the analysis to all ten benchmark topics. For each topic, we use 300 input opinions and examine the minority bias patterns exhibited by different summarization models. Similar to the method in Appendix C.2, we automatically detected minority opinions using embedding-based outlier detection. We then compared the representativeness scores between detected minorities and non-minorities, and conducted a random control experiment by randomly sampling the same number of opinions.

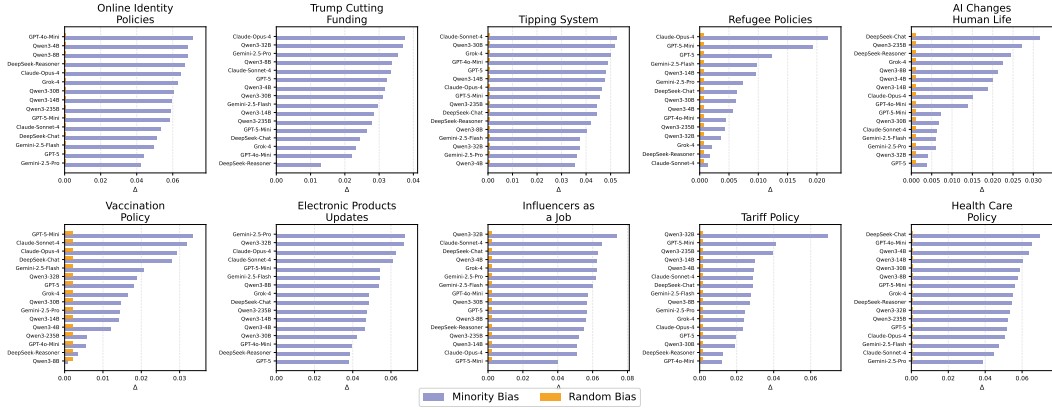

Figure 13: Automatically detected minority bias Vs. random bias across 10 topics (N=300). Each subplot shows the Representativeness Bias of minority opinions relative to non-minority opinions for each model. Purple bars indicate the bias detected using automatic outlier detection (minority bias), while orange bars show the bias from random sampling control (random bias). Across all models and topics, the minority bias (≈ 0.0364) is 34.4× larger than random bias (≈ 0.0011).

Our analysis reveals a systematic and consistent minority bias across all models and topics. The Representativeness Bias of minority opinions relative to non-minority opinions is 34.4× larger than the bias observed in random sampling. The bias is consistent across all models, with minority Bias ranging from 0.0310 to 0.0404, indicating that this is a universal phenomenon rather than model-specific. These findings demonstrate that automatically detected minority opinions systematically

receive lower representativeness scores across diverse topics and models, highlighting a persistent bias in how minority perspectives are represented in summaries.[pDNN]

# D  DETAILS OF MODEL CHOICE

As shown below, we include 18 widely used LLMs, covering both proprietary (closed-source) and open-weight models across several major families: OpenAI GPT, Anthropic Claude, Google Gemini, Alibaba Qwen3, DeepSeek, and xAI Grok. Within the Qwen3 series, `Qwen3-30B-a3b` and `Qwen3-235B-a22b` adopt a Mixture-of-Experts (MoE) architecture, with 3B and 22B active parameters during inference, respectively.

Table 6: List of evaluated summary models, including both API-based frontier systems and open-weight models of varying scales.

| Model | #Size | Form | Creator | Model | #Size | Form | Creator |
|---|---|---|---|---|---|---|---|
| GPT-4o-Mini (Achiam et al., 2023) | N/A | api | OpenAI | Qwen3-0.6B (Team, 2025) | 0.6B | open | Alibaba |
| GPT-5-Mini (OpenAI, 2025) | N/A | api | OpenAI | Qwen3-1.7B (Team, 2025) | 1.7B | open | Alibaba |
| GPT-5 (OpenAI, 2025) | N/A | api | OpenAI | Qwen3-4B (Team, 2025) | 4B | open | Alibaba |
| Claude-4-Sonnet (Anthropic, 2025) | N/A | api | Anthropic | Qwen3-8B (Team, 2025) | 8B | open | Alibaba |
| Claude-4-Opus (Anthropic, 2025) | N/A | api | Anthropic | Qwen3-14B (Team, 2025) | 14B | open | Alibaba |
| Gemini-2.5-Flash (Comanici et al., 2025) | N/A | api | Google | Qwen3-30B-a3b (Team, 2025) | 30B(a3B) | open | Alibaba |
| Gemini-2.5-Pro (Comanici et al., 2025) | N/A | api | Google | Qwen3-32B (Team, 2025) | 32B(a22B) | open | Alibaba |
| DeepSeek-V3.1 (Thinking) (Guo et al., 2025) | 671B | api | DeepSeek | Qwen3-235B-a22b (Team, 2025) | 235B | open | Alibaba |
| DeepSeek-V3.1 (No-Thinking) (Liu et al., 2024) | 671B | api | DeepSeek | Grok-4 (xAI, 2025) | N/A | api | xAI |

# E  DETAILS OF PUBLIC OPINION DATASETS

## E.1  DELIBERATION QUESTIONS.

As shown in Table 7, we construct ten deliberation questions using a two stage process. We first use OpenAI's DeepResearch tool to gather trending public discussion topics from social media and online news, then manually screen out extreme or unsuitable prompts. The final set includes five binary choice policy questions and five open ended societal questions, ensuring coverage of both structured policy debates and broader social issues for summarization and evaluation.[pDNN]

## E.2  WORLDCLOUD.

The wordclouds (see Figure 14) visualize participant opinions, highlighting salient terms and thematic differences across binary and open-ended deliberation topics.

## E.3  DATA QUALITY CHECKING

For all tasks involving human participation (i.e., the collection of the Public Opinion Dataset and the Human Judge Annotations), we conducted systematic quality control. Specifically, we monitored completion times for each participant to filter out responses that were submitted unrealistically quickly, which are indicative of inattentive or low-effort behavior. In addition, we randomized the order of questions for each participant to mitigate potential order bias and ensure fairness in evaluation. All retained data thus reflect responses that passed both attention and fairness checks.

# F  DETAILS OF HUMAN ANNOTATION PROCESS

## F.1  RING-BASED SUMMARY MATCHING ALGORITHM

For the comparison task (see §2.2), we adopt a ring-based matching algorithm (see Alg. 1) rather than random sampling. This ensures balanced and repeated pairing of summaries within each deliberation question.

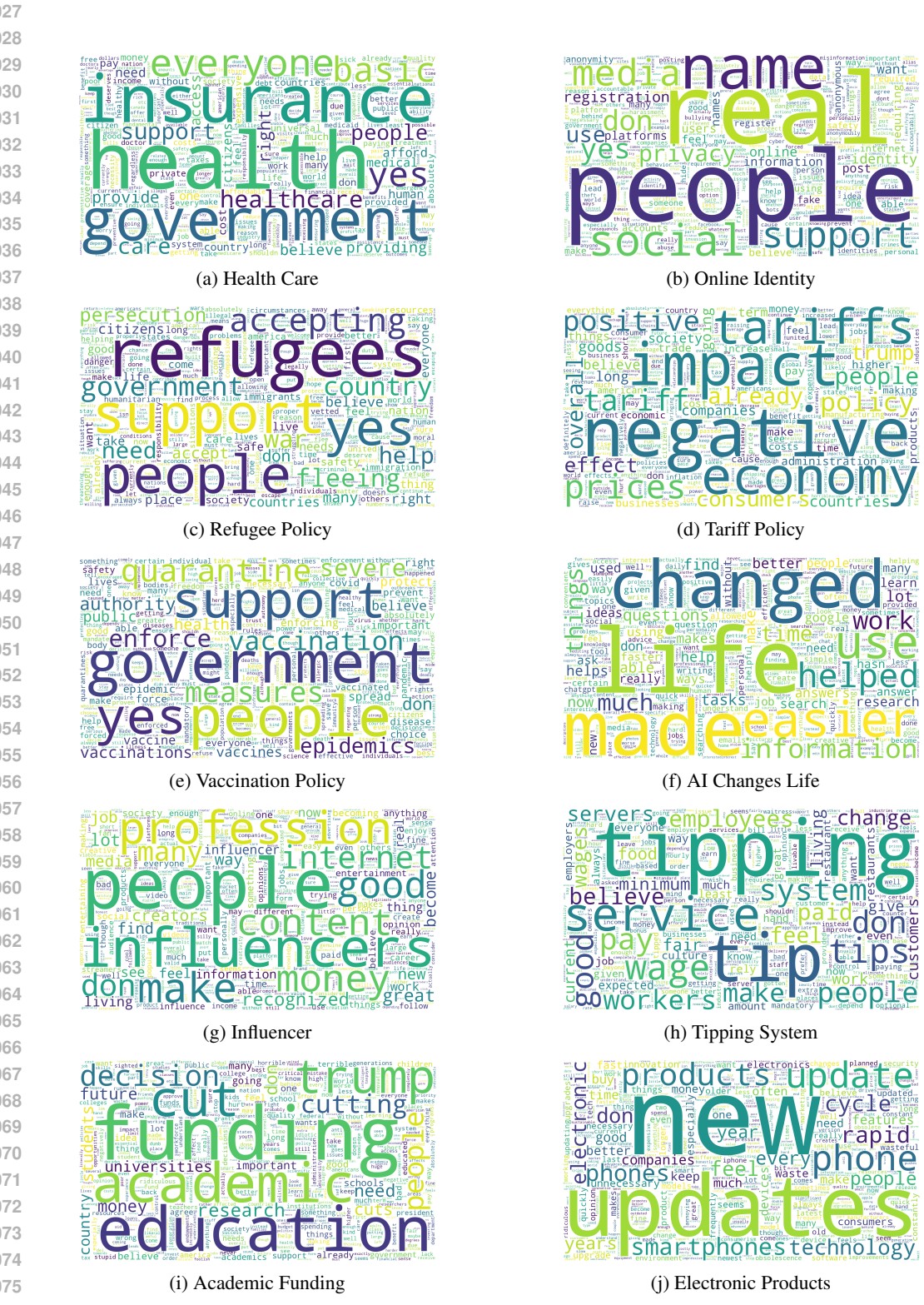

Figure 14: Wordclouds for the annotation dataset across all topics.

Table 7: Deliberation questions.

| Questions | Type | Description |
|---|---|---|
| "Tipping System" | Open-Ended | What is your opinion on tipping, and if given the chance, how would you improve or change the current tipping system? |
| "AI Changes Life" | Open-Ended | How has AI changed your life? |
| "Academic Funding" | Open-Ended | What are your thoughts on Trump's decision to cut academic funding? |
| "Influencer" | Open-Ended | What is your opinion on internet influencers (e.g., streamers, bloggers, short video creators) increasingly becoming a recognized profession? |
| "Electronic Products" | Open-Ended | What is your opinion on the rapid update cycle of electronic products, especially smartphones? |
| "Tariff Policy" | Binary | Do you think the current tariff policy under the Trump administration will have a positive or negative impact on the overall U.S. economy and society? |
| "Health Care" | Binary | Do you support the government provide basic health insurance for everyone? |
| "Vaccination Policy" | Binary | Do you support the government having the authority to enforce vaccination and quarantine measures during severe epidemics? |
| "Refugee Policy" | Binary | Do you support the government accepting more refugees fleeing war or persecution? |
| "Online Identity" | Binary | Do you support requiring real-name registration on social media platforms, where users must register and post under their real identity? |

Pairing is performed independently for each question: summaries are aggregated and randomly permuted before pairing. In the default mode, let $n$ be the number of summaries for a question and $k = \texttt{min\_comparisons\_per\_summary}$ (default $k = 6$). For each index $i \in \{0, \dots, n-1\}$ and each offset $o \in \{1, \dots, k\}$, we generate

$$(A, B) = \left(s_i,\ s_{(i+o) \bmod n}\right).$$

If a total number of pairs $M$ is specified instead, we compute $k = \lfloor M/n \rfloor$ and $r = M \bmod n$, generate all pairs for $o \in \{1, \dots, k\}$ as above, and then add one extra pair with offset $k + 1$ for the first $r$ indices.

## G    DETAILS OF METRICS

### G.1    SUMMARIZATION QUALITY METRIC

We evaluate summary quality along four deliberation-relevant dimensions:

- **Representativeness:** To what extent does the summary reflect the annotator's perspective?
- **Informativeness:** How much useful information does the summary provide?
- **Neutrality:** Does the summary present a balanced and unbiased view of the issue?
- **Policy Approval:** Would the annotator approve of this summary being used by policymakers to make decisions?

**Human annotation.** These four metrics were operationalized in two complementary annotation tasks. As shown in Figure 15, in the *rating task*, each summary was independently scored on a five-point Likert scale (poor, slightly poor, neutral, slightly good, good) for all four dimensions. In the *comparison task*, annotators compared two summaries for the same question and indicated which performed better on each dimension using a five-level ordinal scale.

**Automatic benchmarking.** In our evaluation framework, we benchmark LLMs by comparing their predicted judgments against human annotations. Our model, DELIBERATIONJUDGE, is a DeBERTa-based judge that outputs a four-dimensional score vector in the normalized range $[0, 1]$, where each

---

**Algorithm 1** Ring-Based Summary Matching

---

**Require:** Summaries $S = [s_0, \ldots, s_{n-1}]$, seed, either $k$ or $M$
1: $S \leftarrow \text{PERMUTE}(S, \text{seed}); n \leftarrow |S|$
2: **if** $M$ is specified **then**
3:      $k \leftarrow \lfloor M/n \rfloor; r \leftarrow M \bmod n$
4: **else**
5:      $k \leftarrow \text{min\_comparisons\_per\_summary}$
6: **end if**
7: $Pairs \leftarrow \varnothing$
8: **for** $i = 0$ **to** $n - 1$ **do**
9:      **for** $o = 1$ **to** $k$ **do**
10:         $j \leftarrow (i + o) \bmod n$
11:         append $(s_i, s_j)$ to $Pairs$
12:      **end for**
13: **end for**
14: **if** $M$ is specified **then**
15:      **for** $i = 0$ **to** $r - 1$ **do**
16:         $j \leftarrow (i + (k + 1)) \bmod n$
17:         append $(s_i, s_j)$ to $Pairs$
18:      **end for**
19: **end if**
20: **return** $Pairs$

---

value corresponds to one of the four metrics. These continuous predictions serve as summary-level scores for downstream comparison with human ratings and judgments.

## G.2 ANALYTICAL METRIC

**Relative Preference Score**    For each model $\mathcal{M}$ and question $q \in \mathcal{Q}$, we define

$$\text{Diff}(\mathcal{M}, q) = \frac{1}{4} \sum_{d \in \{\text{rep,inf,neu,pol}\}} \hat{y}_q^{(d)} - \frac{1}{|\mathcal{Q}|} \sum_{q' \in \mathcal{Q}} \frac{1}{4} \sum_{d \in \{\text{rep,inf,neu,pol}\}} \hat{y}_{q'}^{(d)},$$

where $\hat{y}_q^{(d)}$ denotes the score on dimension $d$ for question $q$ produced by judge $\mathcal{J}_\theta$. Positive values indicate above-average performance, while negative values indicate below-average performance.

# H DETAILS OF JUDGE MODEL TRAINING

## H.1 MULTI-OUTPUT REGRESSION MODEL TRAINING

We implement a multi-output regression model to predict quality ratings across four key dimensions: perspective representation, informativeness, neutrality balance, and policy approval. Our approach employs several optimization strategies to enhance correlation performance between predicted and ground truth scores.

### H.1.1 MODEL ARCHITECTURE

The model architecture consists of a pre-trained transformer encoder (DeBERTa-v3-base (He et al., 2020)) followed by a task-specific regression head. The encoder processes the concatenated input sequence containing the question, annotator opinion, and summary text, separated by special [SEP] tokens with descriptive prefixes formatted as "Question: $q_i$ [SEP] Annotator opinion: $o$ [SEP] Summary: $S_{\mathcal{M}, \tilde{\mathcal{O}}_i}$ [SEP]". We extract the [CLS] token representation from the final encoder layer and pass it through a hidden layer that reduces dimensionality by half with GELU activation, followed by dropout regularization for enhanced feature learning. The regression head consists of a linear layer that maps the transformed features to four-dimensional outputs corresponding to the evaluation criteria (perspective representation, informativeness, neutrality balance, and policy approval). To constrain predictions to a valid range, we apply a sigmoid activation function that produces outputs in

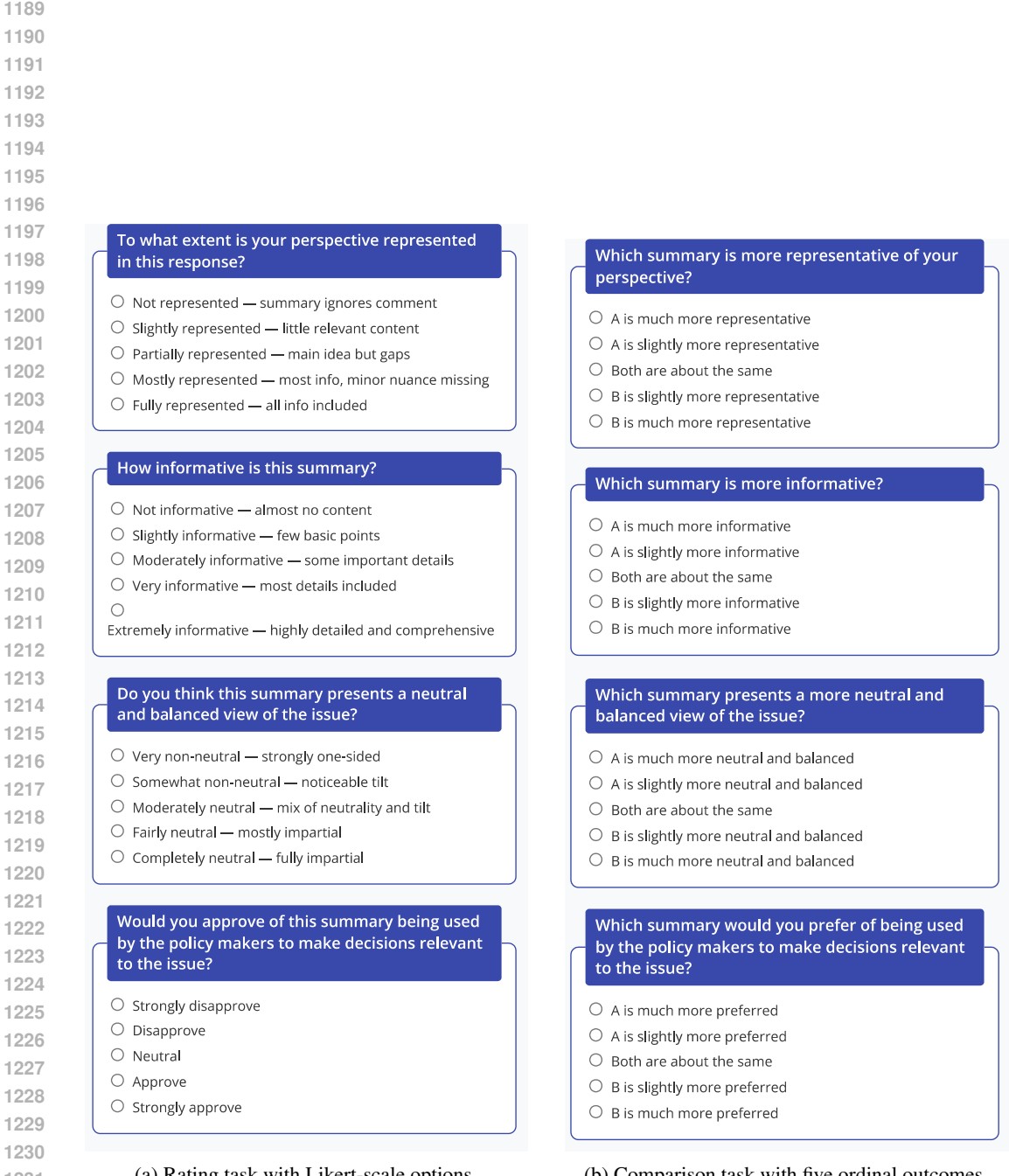

(a) Rating task with Likert-scale options.     (b) Comparison task with five ordinal outcomes.

Figure 15: Screenshots of the annotation interfaces: (a) rating task and (b) comparison task.

the $[0, 1]$ range. Ground truth scores are normalized from their original range $[-1, 7]$ to $[0, 1]$ using min-max normalization ($y = \frac{y^{\text{raw}} - (-1)}{7 - (-1)}$) to improve training stability and convergence.

### H.1.2 TRAINING PARAMETERS

To maximize correlation performance, we incorporate several advanced training techniques and carefully tuned hyperparameters.

**Loss Function and Regularization:** We employ Huber loss with $\delta = 1.0$ instead of standard MSE to enhance robustness against outliers in human annotations. The model includes dropout regularization (rate = 0.1) applied to both the intermediate hidden layer and after the [CLS] token extraction. We apply gradient clipping with L2 norm = 1.0 for training stability.

**Optimization and Learning Rate Scheduling:** We utilize the AdamW optimizer with a linear learning rate scheduler that includes warmup (ratio = 0.15) and weight decay (0.01). The initial learning rate is set to $4 \times 10^{-5}$ with standard Adam hyperparameters ($\beta_1 = 0.9$, $\beta_2 = 0.999$, $\epsilon = 10^{-8}$).

**Training Configuration:** The complete set of training hyperparameters is as follows: maximum sequence length = 4,096 tokens, training batch size = 8, evaluation batch size = 8, number of epochs = 30, and warmup ratio = 0.15. We use mixed precision training (FP16) with gradient accumulation steps = 2 for an effective batch size of 16.

---

**Alpaca Format SFT data**

**Instruction:** We have made a deliberation with many annotators on the issue...
One annotator's opinion: ...
Below is a summary: `[summary content]`

Please evaluate this summary on 4 criteria:

1. Representation...

2. Informativeness...

3. Neutrality...

4. Approval...

- - - - - - - - - - - - - - - - - - - - - - - - - - - - - - - - - - - - - - - - - - -

**Output:**

```
{
    "perspective_representation": 5,
    "informativeness": 5,
    "neutrality_balance": 5,
    "policy_approval": 5
}
```

---

### H.2 COMPARISON OF DIFFERENT BASE MODELS

We have trained additional encoder-based models and large language models with different scales for comparison. For encoder-based models, we trained DeBERTa-v3-large and Longformer-base-4096 (Beltagy et al., 2020) using the same dataset with hyperparameter sweeping to identify optimal training configurations. These models were specifically chosen for their ability to process longer sequences than other BERT-like models. For LLMs, we implemented two distinct training approaches. The first approach follows the same regression framework as the encoder-based models, where we adapt Qwen3-0.6B and Qwen3-4B for multi-output regression by adding a custom regression head on top of the pre-trained causal language model. This regression head employs a multi-layer architecture that progressively reduces dimensionality using LayerNorm, GELU activation, and dropout for regularization. The second approach leverages the instruction-following capabilities of LLMs by constructing an alpaca-format SFT dataset (Ding et al., 2023) based on the annotation data. In this approach, we directly fine-tune Qwen3-4B to generate structured JSON-formatted scores as text output, treating the regression task as a text generation problem rather than a traditional regression objective.

Table 8 presents a comparison of results across different models and training strategies. Several key insights emerge from this comparison. DeBERTa-v3-base with regression training consistently outperforms other models across all evaluation dimensions, achieving the highest correlations. This supports our core approach of employing encoder-based models with task-specific regression heads for multi-dimensional summary evaluation. Interestingly, increasing model size does not necessarily lead to better performance: DeBERTa-v3-large underperforms the base model across all dimensions, indicating that the base model achieves an optimal balance between capacity and generalization for this task.

Table 8: Performance comparison across different models and training strategies. Results show Pearson and Spearman correlation coefficients for each evaluation dimension.

| Model | Representativeness | | Informativeness | | Neutrality | | Policy Approval | |
|---|---|---|---|---|---|---|---|---|
| | Pearson | Spearman | Pearson | Spearman | Pearson | Spearman | Pearson | Spearman |
| DeBERTa-v3-base (Regression) | **0.504** | **0.470** | **0.454** | **0.444** | **0.492** | **0.492** | **0.416** | **0.381** |
| DeBERTa-v3-large (Regression) | 0.159 | 0.162 | 0.221 | 0.203 | 0.129 | 0.125 | 0.174 | 0.162 |
| Longformer-base-4096 (Regression) | 0.097 | 0.098 | 0.209 | 0.219 | 0.158 | 0.156 | 0.127 | 0.125 |
| Qwen3-0.6B (Regression) | 0.125 | 0.136 | 0.231 | 0.249 | 0.210 | 0.205 | 0.196 | 0.186 |
| Qwen3-4B (Regression) | 0.191 | 0.197 | 0.215 | 0.218 | 0.215 | 0.207 | 0.189 | 0.188 |
| Qwen3-4B (SFT) | 0.338 | 0.289 | 0.153 | 0.157 | 0.211 | 0.188 | 0.289 | 0.244 |

Beyond the supervised model comparison, our few-shot evaluation with frontier LLMs further highlights the strength of DeliberationJudge. As shown in Table 9, DeliberationJudge surpasses all few-shot prompting baselines by a clear margin across every evaluation dimension. Even with 5-shot demonstrations, strong models such as GPT-5, GPT-5-mini, and Claude-3.7-Sonnet fall short of the correlations achieved by our DELIBERATIONJUDGE. [Dtym]

Table 9: Performance comparison across our DELIBERATIONJUDGE Model and Few-shot baselines with strong LLMs.[Dtym]

| Model | Representativeness | | Informativeness | | Neutrality | | Policy Approval | |
|---|---|---|---|---|---|---|---|---|
| | Pearson | Spearman | Pearson | Spearman | Pearson | Spearman | Pearson | Spearman |
| DeliberationJudge | **0.504** | **0.470** | **0.454** | **0.444** | **0.492** | **0.492** | **0.416** | **0.381** |
| GPT-5 3-shot | 0.398 | 0.321 | 0.143 | 0.155 | 0.245 | 0.214 | 0.360 | 0.306 |
| GPT-5 5-shot | 0.422 | 0.372 | 0.116 | 0.131 | 0.226 | 0.205 | 0.319 | 0.269 |
| GPT-5-mini 3-shot | 0.399 | 0.347 | 0.094 | 0.095 | 0.241 | 0.221 | 0.344 | 0.293 |
| GPT-5-mini 5-shot | 0.399 | 0.336 | 0.103 | 0.104 | 0.210 | 0.187 | 0.305 | 0.268 |
| Claude-3.7-sonnet 3-shot | 0.372 | 0.302 | 0.145 | 0.157 | 0.320 | 0.278 | 0.353 | 0.304 |
| Claude-3.7-sonnet 5-shot | 0.365 | 0.309 | 0.164 | 0.176 | 0.309 | 0.271 | 0.348 | 0.307 |

## H.3 OUT-OF-DISTRIBUTION TEST

To evaluate the robustness of DELIBERATIONJUDGE under out-of-distribution (OOD) conditions, we introduce two new deliberation questions that are not included in the original ten topics:

- **School Cellphone Use** (Binary): *"Do you support banning students from using personal cellphones during school hours?"*
- **Workplace Flexibility** (Open-ended): *"What are your thoughts on the rise of flexible work arrangements, such as remote or hybrid work, and how should workplaces adapt?"*

For each question, we repeat the full Stage 1 to Stage 3 pipeline. Specifically, for each question, we collect 100 new opinions for summary generation and recruit an additional 100 annotators to provide rating and comparison judgments using the same mapping:

$$\Phi : (q_i, o_S^{(a)}, S, S') \mapsto (r(q_i, S), c(q_i, S, S')).$$

We then apply DELIBERATIONJUDGE to these two OOD test sets and compute Spearman correlations with human judgments using the same evaluation criteria as in the main benchmark. The correlations remain positive but are noticeably lower than in-domain results, indicating that while DELIBERATIONJUDGE transfers partially to unseen deliberation settings, strong generalization to

new topics remains challenging. These findings are consistent with the known data requirements of human preference modeling and highlight the importance of larger and more diverse deliberation datasets for future work.

Table 10: Spearman correlations of DELIBERATIONJUDGE on out-of-distribution datasets.

| Question | Represent. | Inform. | Neutral. | Policy |
|---|---|---|---|---|
| In Domain | 0.47 | 0.44 | 0.49 | 0.38 |
| OOD (School Cellphone Use & Workplace Flexibility) | 0.20 | 0.05 | 0.14 | 0.23 |

We further test whether a stronger base model can close the transfer gap by fine tuning a GPT-4.1 model on training set $\mathcal{T}_{\text{train}}$ from the 10 in domain topics with same setting when train DELIBER-ATIONJUDGE. The model was trained only on these original topics and then evaluated on both in-domain testset $\mathcal{T}_{\text{test}}$ and OOD testset introduced above. As shown in Table 11, when evaluated on the in domain test set, the fine tuned model showed consistent positive correlations across all dimensions. When tested on the two OOD questions, the performance dropped by approximately 2 to 30 times compared with the in domain test sets. This sharp drop shows that even a high capacity model trained on all available in domain data cannot generalize preference signals to new topics.[wcNP]

Table 11: Spearman correlations of GPT-4.1 fine tuned on the in domain topics, evaluated on the in domain test sets and the two OOD datasets.

| Test Set | Represent. | Inform. | Neutral. | Policy |
|---|---|---|---|---|
| In Domain | 0.29 | 0.15 | 0.12 | 0.18 |
| OOD (School Cellphone Use & Workplace Flexibility) | 0.16 | −0.004 | 0.02 | 0.06 |

## I  DETAILS OF EXPERIMENT

### I.1  PROMPTS

---
**LLM Summarization Prompt**

**User:** In each line, I provide you with human comments for a deliberation question `{question}`. At the end, generate an overall summary of the comments. Please do not mention the total number of comments or participants. If you need to provide statistical information, use percentages instead of absolute numbers.
Here are the comments:
`{comments}`

---

## J  DETAILS OF HUMAN DATA COLLECTION

All human-involved procedures were implemented using two annotation platforms, `potato` (Pei et al., 2022) and `deliberation.io` (Pei et al., 2024), with participant recruitment conducted via Prolific[1]. The following screenshots illustrate the user interface and the design of annotation questions for each stage of human comment collection and evaluation.

### J.1  PUBLIC OPINION DATA COLLECTION

For the ten questions introduced in Table 7, we launched ten separate studies on Prolific, each corresponding to one question. In each study, we recruit participants from the United States, aiming for a representative population across demographic groups (see §L for details). Moreover, Figures 16-19 illustrate the full opinion-submission workflow for the *tipping system* study, which we present as a

---
[1] https://www.prolific.com/

pipeline example. In addition, we show in Figure 20 the Deliberation.io management interface that we used to coordinate the collection of all 10 studies.

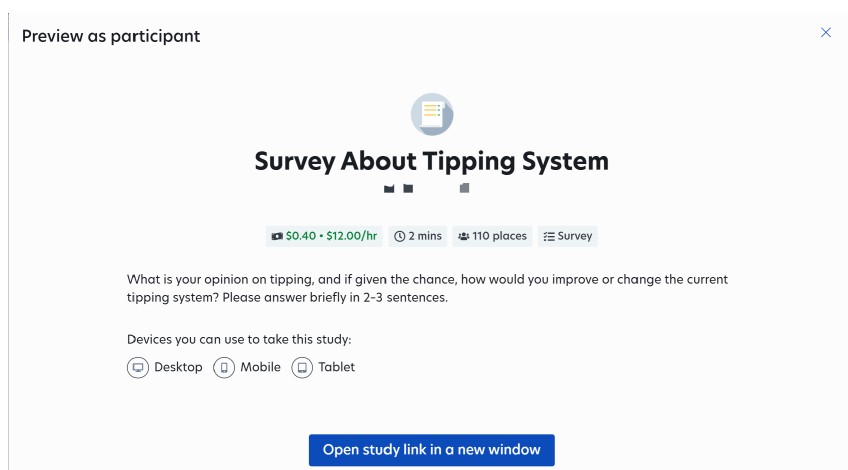

Figure 16: Example of Public opinion collection recuitment page powered by Prolific

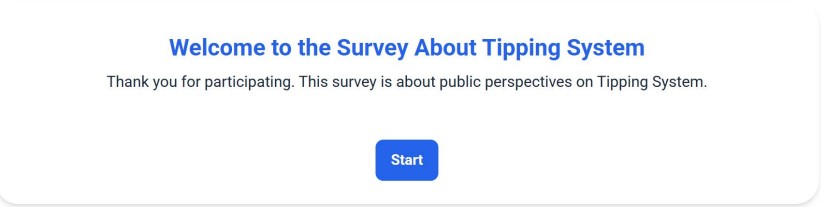

Figure 17: Example of public opinion collection start page powered by Deliberation.io

## J.2 HUMAN JUDGE DATA COLLECTION

To collect human judgment data, we designed a structured annotation pipeline implemented on the `potato` platform (Pei et al., 2022). Recruitment of annotators was conducted via Prolific, ensuring a pool of participants distinct from those who contributed to the public opinion dataset. Each annotator was guided through a standardized workflow: recruitment and consent, task instructions, writing their own opinion, rating a model-generated summary along four evaluation dimensions, and final submission. Figures 21–26 provide interaction interface screenshots of this process.

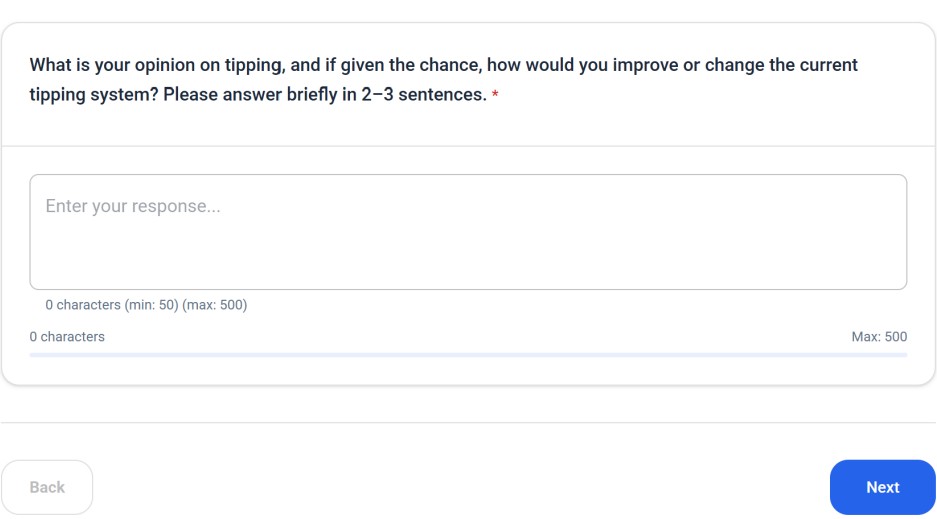

Figure 18: Example of public opinion collection question page powered by Deliberation.io

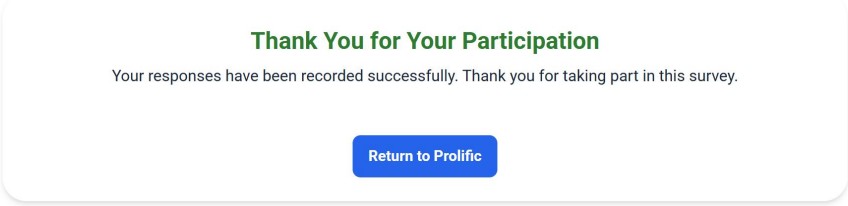

Figure 19: Example of public opinion collection ending page powered by Deliberation.io

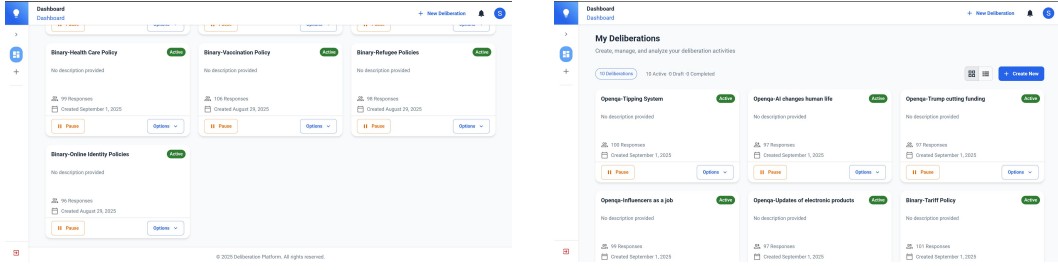

Figure 20: Example of the public opinion collection management interface on Deliberation.io.

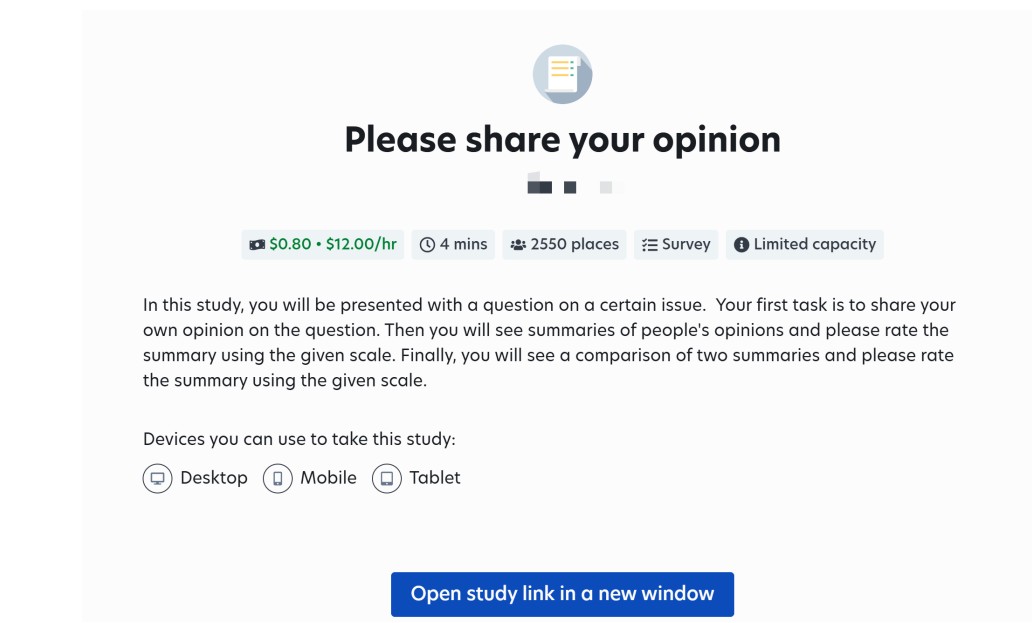

Figure 21: Recruitment page for human judge annotation (via Prolific).

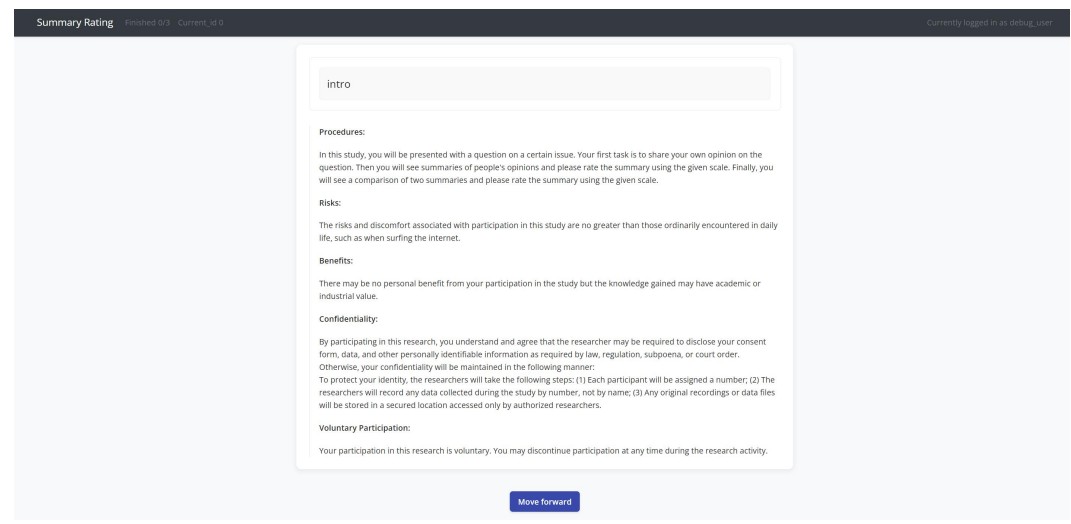

Figure 22: Task introduction page in the potato annotation system.

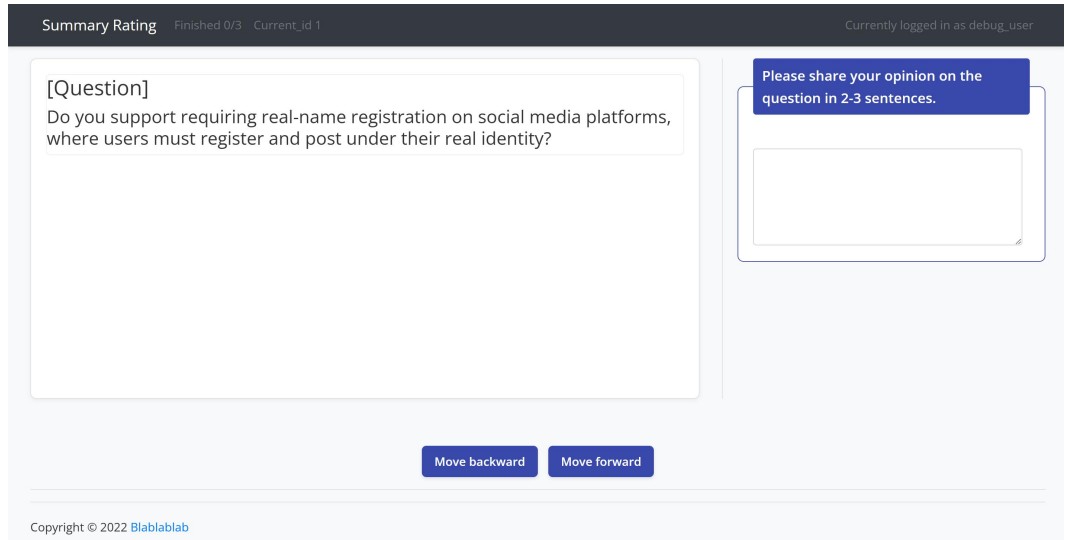

Figure 23: Annotator writing their own opinion for the given deliberation question.

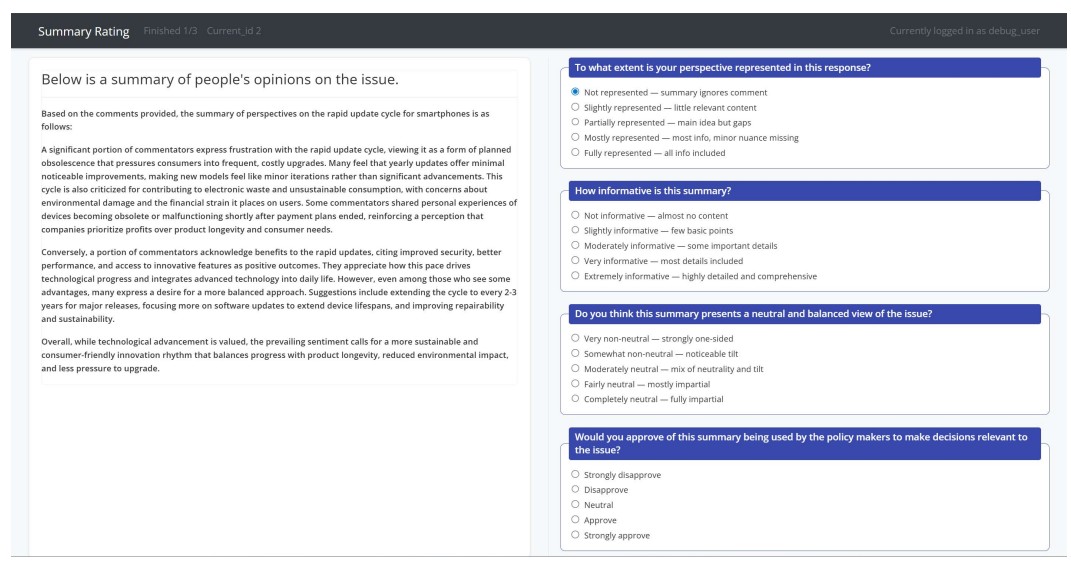

Figure 24: Annotation interface for rating judgment task.

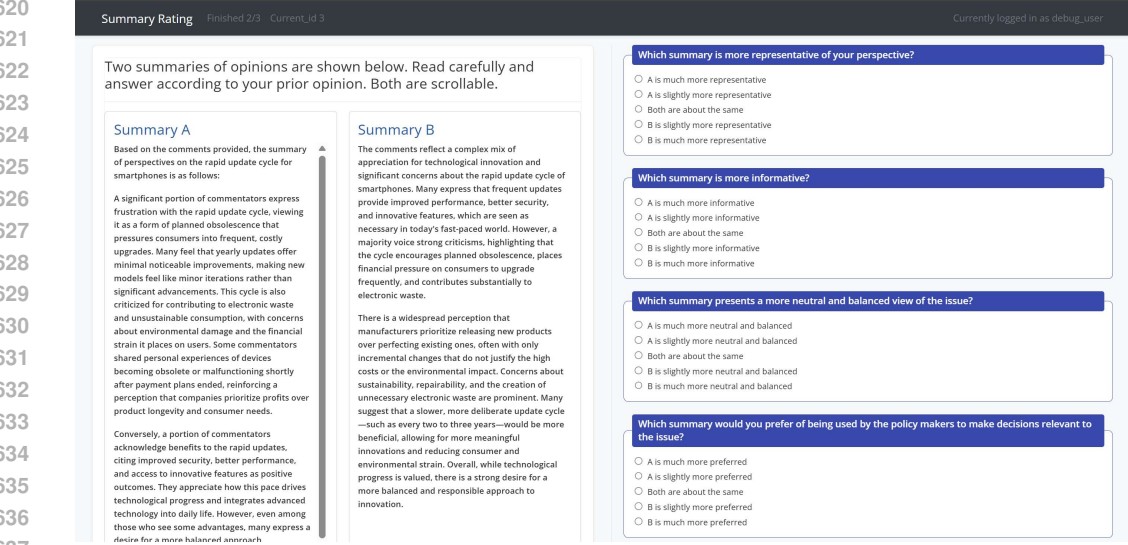

Figure 25: Annotation interface for compare judgment task.

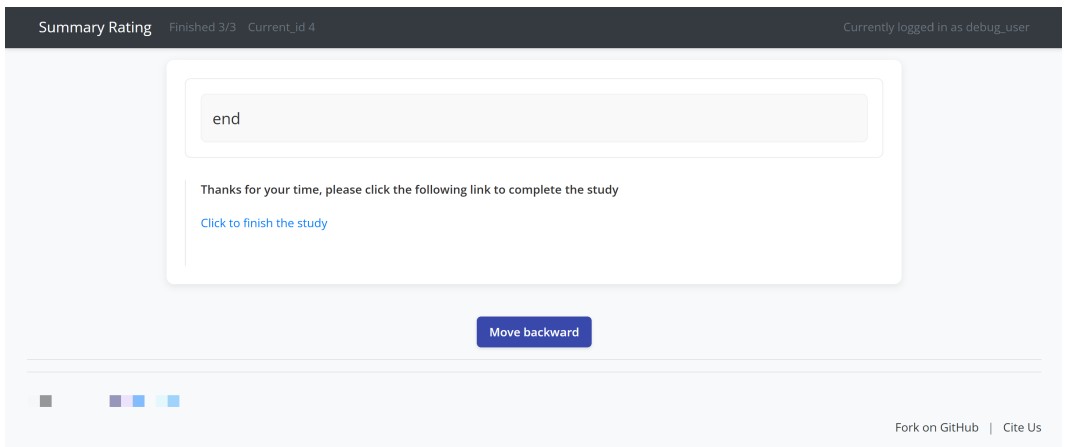

Figure 26: Final submission and completion page in the potato system.

## K    ADDITIONAL RESULTS

### K.1    HUMAN ANNOTATIONS

In this subsection, we present the detailed analysis of the human annotation results, as summarized in Figures 27–31. Figure 27 shows the distribution of overall ratings on a 5-point scale, while Figure 28 reports the corresponding distribution in the binary comparison setting. To examine the alignment between rating schemes and the binary comparison setting, Figure 29 illustrates the relationship between rating label distributions and sample-level win rates. Figure 30 further analyzes the impact of the number involved in summary on win rates, suggesting potential position or length biases in human judgments. Finally, Figure 31 presents the correlations among the four evaluated dimensions

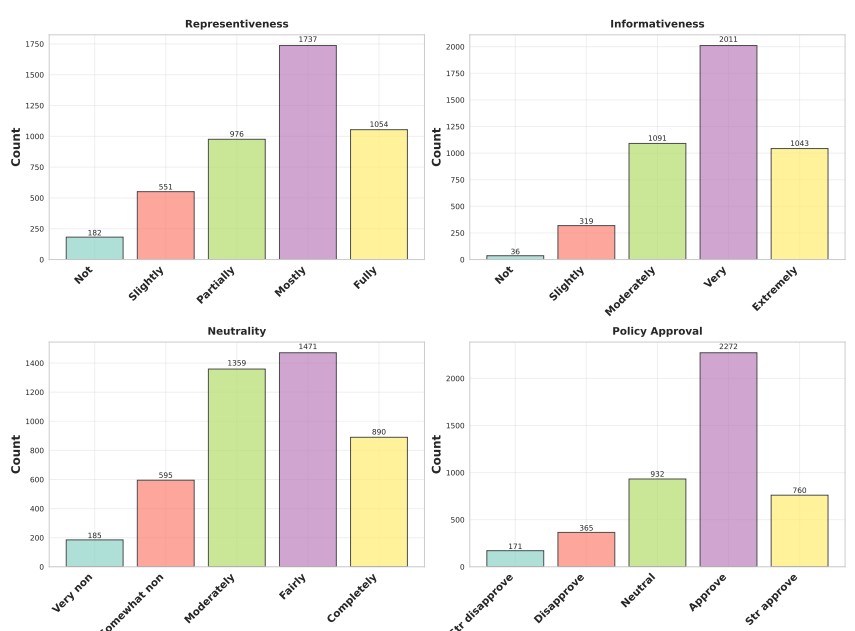

Figure 27: Distribution of overall human ratings on a 5-point scale.

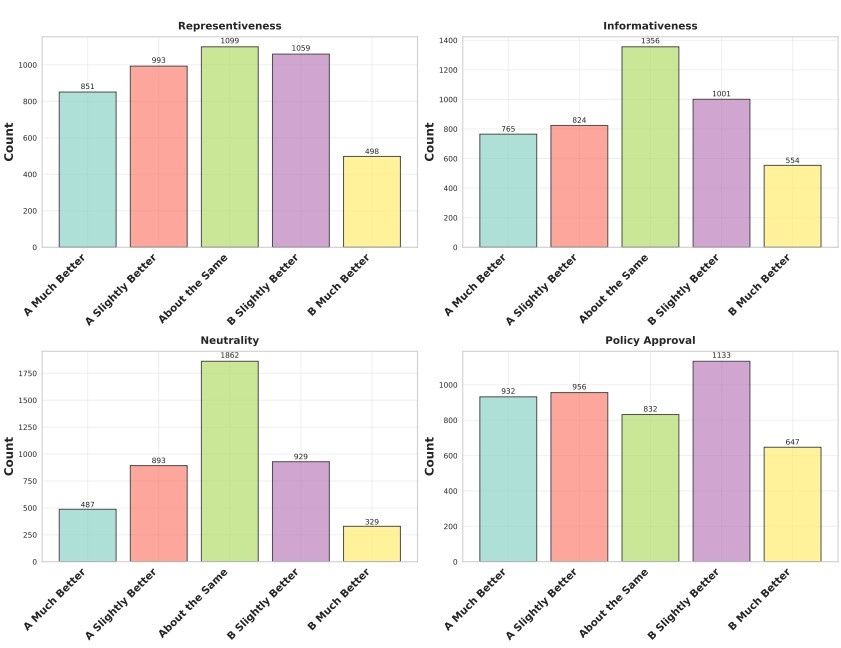

Figure 28: Distribution of human annotations in the binary comparison setting.

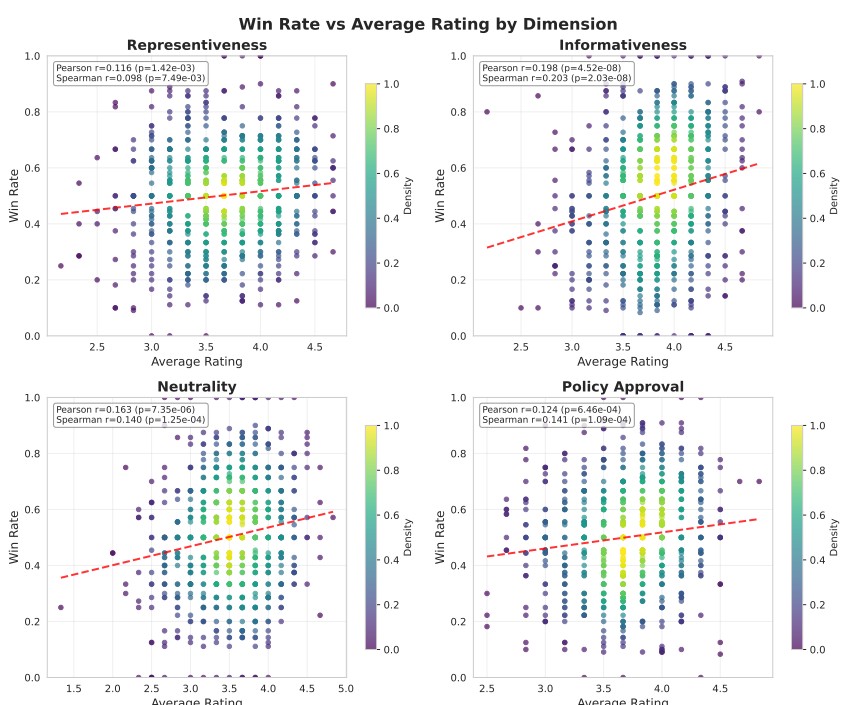

Figure 29: Relationship between rating label distribution and win rate at the sample level.

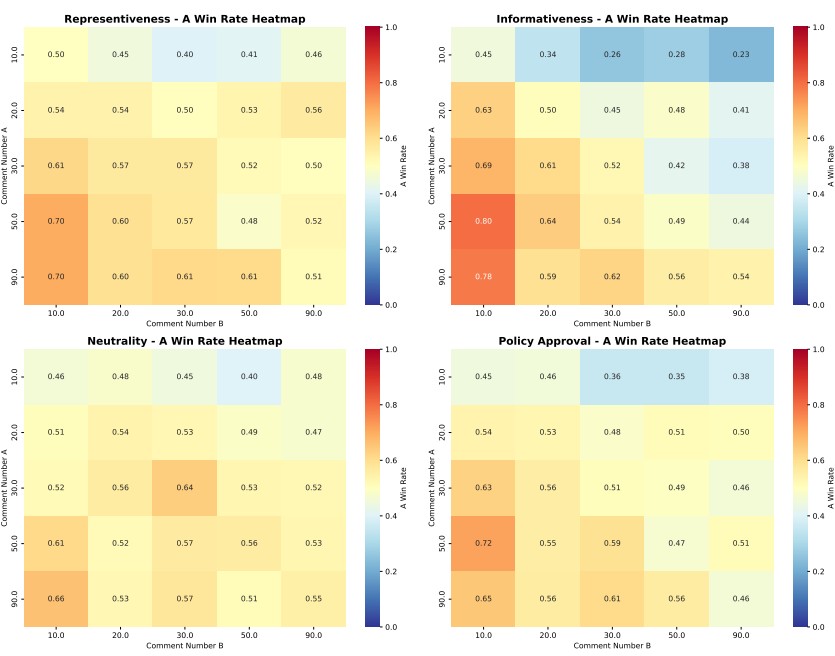

Figure 30: Heatmap of the relationship between the number of comments in a summary and its win rate in the human annotation dataset.

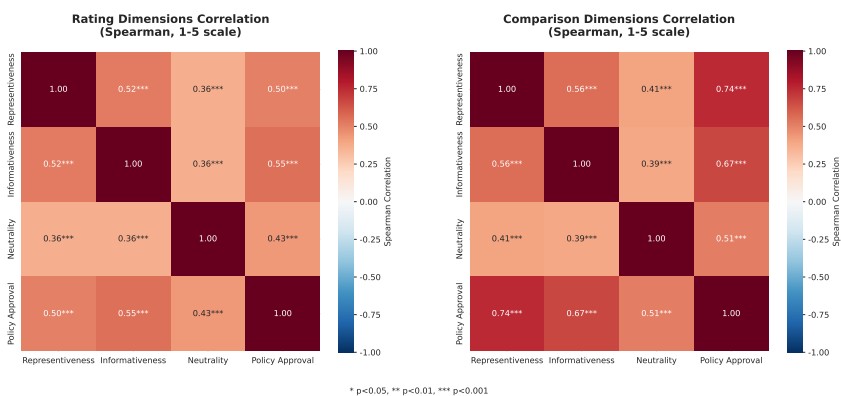

Figure 31: Correlations among the four evaluated dimensions.

## L  HUMAN PARTICIPANTS DEMOGRAPHIC

### L.1  DELIBERATIONBANK: PUBLIC OPINION PART

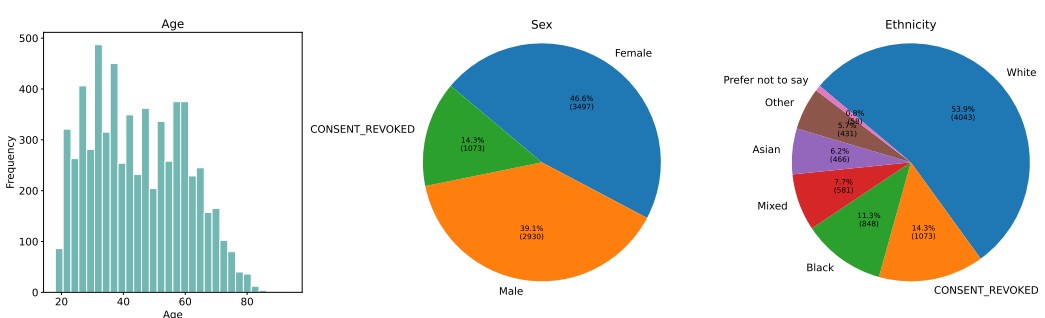

Figure 32: participants demographics for giving public opinion sample (n=7500), including 3000 sample from §2.2 and 4500 sample from §3: age distribution, sex and ethnicity breakdown; pie labels show % and counts.

### L.2  DELIBERATIONBANK: HUMAN JUDGE ANNOTATION PART

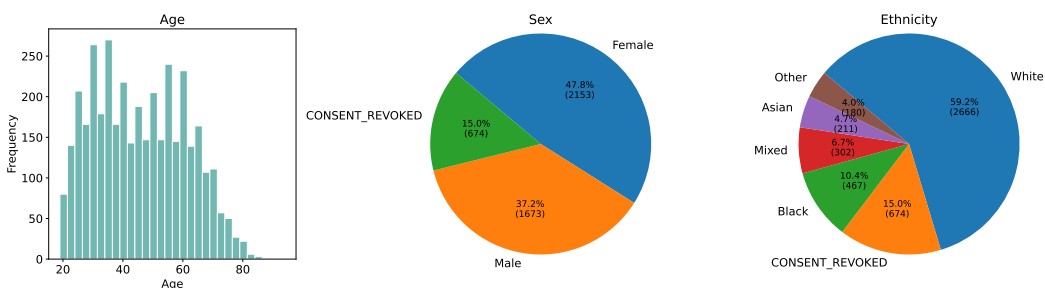

Figure 33: Human annotation participant demographics (n=4500): age histogram; sex and ethnicity pies; labels show % and counts.

## M  THE USAGE OF LARGE LANGUAGE MODELS (LLMS)

LLMs were used only occasionally to help polish the writing (propose new words, grammar, and spelling correction). All technical ideas, experimental designs, analyses, conclusions, and writing

were developed and carried out entirely by the authors. The authors have full responsibility for the final text.

