# OpenReview forum: "Can AI Truly Represent Your Voice in Deliberations? A Comprehensive Study of Large-Scale Opinion Aggregation with LLMs"
_ICLR.cc/2026/Conference — ICLR 2026 Conference Withdrawn Submission_

### Official Review · Reviewer_pDNN · 2025-10-24

**Soundness:** 2
**Presentation:** 3
**Contribution:** 3
**Rating:** 6
**Confidence:** 3

**Summary:**

LLMs should theoretically be capable of supporting deliberation by summarizing discussions in ways that are appropriate for use by policymakers, but no large-scale evaluations exist to benchmark their ability to do so. In response to this, the authors introduce DeliberationBank, a large-scale human-grounded dataset of crowdworker opinions and summary evaluations. They then train automatic evaluators on this dataset, finding that they comfortably outperform zero-shot LLM-as-a-judge, and use it to assess LLM summarization performance.

**Strengths:**

- The problem space is well-motivated, and their contributions (the dataset DeliberationBank and the DeliberationJudge model) are highly useful contributions to this subfield. In particular, the large-scale data collection will be very useful once published.
- Well-designed experiments make a strong case that off-the-shelf LLMs are insufficient as-is for the summarization task due to limited neutrality and representativeness. These are supplemented by detailed order analysis.
- Thorough ablations for the judge design - it’s interesting that DeBERTa does better even than more recent LLMs.

**Weaknesses:**

- Limited discussion of interannotator agreement when collecting human judgments. Many of the annotation dimensions seem highly subjective, and it would be useful to verify that there is high IAA between crowdworker judgments to verify that their judgments can be used as ground-truth values for the dataset. For example, it seems plausible that crowdworkers would have limited understanding of what kind of summaries would be useful for policymakers.
- The minority representation case study seems pretty limited - only running on two topics, given the high inter-topic variance cited in Section 4.2, seems like it may not give the full picture. I’m also unconvinced that self-reports are the best way to classify opinions as minority or non-minority, as participants may not be well-calibrated about others’ opinions. The authors note significant past work on extracting classes of opinions from deliberation datasets - why wouldn’t a similar automatic approach work here?

**Questions:**

- How did you choose the full spectrum of topics in Appendix B1?
- Why is Spearman’s used in some places and Pearson’s in others?
- How do you envision others leveraging your dataset and model in future work?

---

> ### Author Response · Authors · 2025-11-21
>
> Thank you very much for your thoughtful feedback. We have revised the corresponding parts and indicated all updates in **Red** in the new version.
>
> > W1. Concerns About the Reliability of Human Annotations
>
> In our setup, each annoatated instance in $\mathcal{T}_{\mathrm{annotate}}$ corresponds to a specific deliberation participant who first states their own opinion and then rates summaries relative to that personal perspective. Each instance is therefore annotated by a single individual, and conventional IAA metrics comparing multiple annotators on the same item do not apply. This is especially true for dimensions such as representativeness, which reflect an individual’s own viewpoint rather than an objective property of the text; disagreement across individuals would reflect genuine perspective differences rather than annotation noise.
>
> This design choice is consistent with prior work on crowdsourced labeling. Sheng et al. (2008) show that, under fixed annotation budgets, repeatedly labeling the same instance yields quickly diminishing returns and that allocating resources to additional singly labeled items can improve overall data quality. Snow et al. (2008) demonstrate that large, diverse crowds with clear guidelines and basic quality checks can produce aggregate labels that closely approximate expert judgments, supporting the use of a broad, representative annotator pool. In information retrieval, MacAvaney and Soldaini (2023) further show that one shot labeling, where each item receives a single human judgment, can still yield stable evaluation when coupled with appropriate sampling and quality control. We add this reference in our annotation process as an explaination in **lines 161-164**.
>
> Finally, although annotators are not policy experts, the evaluation criteria focus on observable textual properties, whether the summary reflects the annotator’s view, conveys key information, and maintains neutrality, rather than domain knowledge. These properties can be assessed reliably by non-experts. Together, these factors provide a meaningful and reliable supervision signal for training and evaluating DELIBERATIONJUDGE, even though traditional IAA cannot be computed in our single-annotator setting.
>
> ```
> [1] Sheng, Victor S., Foster Provost, and Panagiotis G. Ipeirotis. "Get another label? improving data quality and data mining using multiple, noisy labelers." Proceedings of the 14th ACM SIGKDD international conference on Knowledge discovery and data mining. 2008.
>
> [2] Snow, Rion, et al. "Cheap and fast–but is it good? evaluating non-expert annotations for natural language tasks." Proceedings of the 2008 conference on empirical methods in natural language processing. 2008.
>
> [3]MacAvaney, Sean, and Luca Soldaini. "One-shot labeling for automatic relevance estimation." Proceedings of the 46th International ACM SIGIR Conference on Research and Development in Information Retrieval. 2023.
> ```
>
> > W2. Limitation of minority representation case study
>
> We agree that an automatic approach offers a useful complementary perspective. In the revised version (**Section 5.2 Objectively Defined Minority**), we include an embedding-based minority detection that clusters opinions by semantic similarity and compares the resulting minority group against randomly sampled data. We find that automatically detected minorities still exhibit substantially lower representativeness which consistent with self-reported method, confirming that the bias persists under an objective definition.
>
> Moreover, to address your concern on limited topic analysis, **we extend the automatic minority detection across all 10 topics** to examine whether the pattern generalizes beyond the two case study questions. As detailed in **Appendix C.4**, every topic shows the same directional gap: automatically detected minority opinions receive consistently lower representativeness scores than non minority groups, with effect sizes comparable to those observed in the main experiments. This cross topic consistency indicates that the minority bias is not an artifact of a specific question, but a systematic property of current LLM summarizers. We refer this new contribution in **line 528**.
>
> ### **Continue Below**

---

> > ### Author Response · Authors · 2025-11-21
> >
> > > Q1. How to construct the topic list
> >
> > We constructed the topic set using a two stage process. First, we used OpenAI’s DeepSearch tool to gather a broad pool of trending public discussion questions by scanning high volume social media (e.g. X, Reddit) and online discourse (e.g. Forbes). This gave us a wide range of issues that people actively debate. Second, we performed manual screening to ensure suitability for deliberation research. We removed extreme sensitive contents and selected topics that exhibit clear viewpoint diversity, cover both open ended and binary formats, and span domains such as technology, public health, economic policy, and social values. The final ten topics represent a balanced set of trending, diverse, and substantively meaningful public deliberation questions. In the revised version, we add a detailed description of this collection procedure in **Appendix E.1**. We highlight this reference in **line 144**.
> >
> > > Q2. Why use different correlation?
> >
> > All four evaluation dimensions use a 1–5 ordinal scale, so the appropriate primary measure of correlation is Spearman, which assesses whether the ranking induced by model predictions aligns with the ranking in human judgments rather than assuming a linear relationship. Pearson is included only as an auxiliary metric for completeness. In the revised version, we update the left panel of **Figure 4 (Left)** to report Spearman correlation accordingly, while keeping the rest of the analysis unchanged.
> >
> > > Q3. How do you envision others leveraging your dataset and model in future work?
> >
> > The first intended use of DELIBERATIONJUDGE is as an evaluation component for assessing whether LLMs can summarize deliberation content in a way that preserves representativeness, informativeness, neutrality, and approval. This enables rigorous and transparent measurement of model performance in deliberation focused summarization, which is essential for policy relevant applications. Beyond evaluation, the judge model can also serve as a reward model for training or fine tuning LLMs toward socially aware, diversity sensitive summarization behaviors.
> >
> > Our dataset, DeliberationBank, likewise provides substantial value for future research. It contains 7,500 human written opinions and annotation across ten questions, forming a large scale, clean, and structurally unified corpus grounded in real participant perspectives. Such a dataset enables multiple lines of work, including model alignment, bias and minority representation analysis, robustness evaluation, diversity aware summarization, and training new judgment or reward models. In the revised version, we also add a dedicated in **Appendix A** that discusses these directions explicitly. We edit the **line 329** to show this appendix reference.

---

> > > ### Author Response · Authors · 2025-11-26
> > >
> > > Dear reviewer,
> > >
> > > We hope that the clarifications above  included in the revised version have addressed your concerns. If the revisions meet your expectations, we would appreciate your consideration of updating the score to reflect the new results and discussion. Please feel free to let us know if any points remain unclear!
> > >
> > > best,
> > > authors

---

> > > > ### Comment · Reviewer_pDNN · 2025-11-26
> > > >
> > > > I thank the authors for their detailed response to my review and hope that they found my comments useful.
> > > >
> > > > I remain somewhat skeptical of the reliability of human annotations. I agree that individual human annotators can still yield appropriate labels on certain tasks when there is sufficient quality control, and that some variance in annotations is due to subjective disagreements rather than just noise. However, given that the benchmark is built directly on top of the annotations, I think it's important to conduct some sanity checks of annotation quality on top of simply filtering out highly inattentive annotators. This is especially the case for dimensions like policymaker utility, which can be decomposed into individual criteria, but remain relatively complex annotation tasks that require aggregating several intermediate judgments together.
> > > >
> > > > I appreciate the work the authors did on improving their minority representation analysis and answering my other questions, which has definitely raised my assessment of the paper. However, given my other uncertainties, I still think maintaining my score is more reflective of my assessment than increasing it to 8.

---

### Official Review · Reviewer_wcNP · 2025-10-31

**Soundness:** 2
**Presentation:** 3
**Contribution:** 3
**Rating:** 4
**Confidence:** 3

**Summary:**

This paper presents DELIBERATIONBANK, a large-scale dataset for studying large language models (LLMs) in the context of public deliberation summarisation, and introduces DELIBERATIONJUDGE, a fine-tuned DeBERTa model designed to evaluate summaries across four human-centered dimensions: representativeness, informativeness, neutrality, and policy approval. The dataset combines 3,000 free-form opinions from ten societal deliberation topics with 4,500 human annotations rating summaries generated by 18 different LLMs. Using this benchmark, the authors analyse systematic weaknesses in deliberation summarisation (e.g., underrepresentation of minority perspectives, input-order sensitivity) and demonstrate that their fine-tuned judge model achieves higher correlation with human judgments and greater efficiency compared to general-purpose LLM evaluators.

**Strengths:**

* The creation of DELIBERATIONBANK fills an important gap in large-scale, human-grounded evaluation of deliberation summarisation, a topic with high social and policy relevance.
* The evaluation across 18 LLMs, multiple topics, and controlled input scales provides a thorough empirical picture of current model capabilities and weaknesses.
- The study explicitly examines minority opinion coverage, a dimension rarely addressed in summarisation benchmarks, adding ethical and societal depth.
* The human evaluation design (rating + comparison tasks) is clear, systematic, and well-documented, yielding high-quality supervision data.
- The paper is well-written and easy to follow, with clear figures and a well-structured argument.

**Weaknesses:**

* The main modelling component, DELIBERATIONJUDGE, is a straightforward fine-tuning of DeBERTa with a regression head and Huber loss. While practical, it introduces no methodological innovation beyond supervised fine-tuning. It would be good if the authors could highlight generalisable technical components of their work that might be useful for the community.

- The train/test split (random 80/20) likely leads to substantial overlap in topics, question types, and summarisation styles, so the model’s generalisation to unseen deliberations or unseen LLM summarisers is unclear. Other political debate datasets such as X-Stance (Vamvas et al. 2020) make clear distinctions between topics and questions, i.e., topics in train are not in test. This does not become apparent from this work.

- The study identifies fairness and minority-representation gaps but does not probe why these biases arise or how to mitigate them, which limits the conceptual insight.

-  The use of a fine-tuned DeBERTa judge resembles prior “LLM-as-a-judge” work, differing mainly by domain rather than by technique.

**Questions:**

1. How distinct are the train and test sets in terms of deliberation topics and summarisation models? Can the authors show that the topics handled in train and test are genuinely different and train is not leaking information to test?

2. Did the authors evaluate performance on held-out questions or unseen summarisers to test generalisation beyond the training distribution?

3. Could the approach be extended to explicitly model deliberative diversity (e.g., stance-conditioned evaluation or contrastive objectives)?

4. What interpretability analyses, if any, were conducted to understand what linguistic or semantic cues DeliberationJudge relies on?

5. Beyond efficiency, how does the judge perform when used for ranking or selection of summaries, rather than scoring them in isolation?



I find this paper interesting and valuable as a dataset and benchmark contribution, but technically limited in terms of modelling innovation. The empirical analysis is careful and socially relevant, but the methodological core (a DeBERTa fine-tune) does not offer new ideas or generalisable techniques. I recognise the value of a deliberation dataset, therefore I would currently rate it as **4 (Borderline / Reject)** but would increase my score if the authors strengthen the framing as a dataset/benchmark paper and clarify the out-of-distribution evaluation.

---

> ### Author Response · Authors · 2025-11-21
>
> Thank you for raising these questions. We have revised our paper accordingly and highlighted the modifications in **Blue**.
>
> > W1. highlight the contribution to the community
>
> Our main motivation is not to introduce a new model architecture, but to establish a reliable and scalable evaluation framework for large-scale deliberation summarization. The methodologically valuable contribution lies in the generalizable evaluation pipeline we develop as we present in **line 100-114**. Specifically:
>
> - A large-scale, human-grounded benchmark dataset (DELIBERATIONBANK): we construct a dataset involving 7,500 participants from a U.S.-representative panel, consisting of (i) 3,000 free-form public opinions across 10 societal questions, and (ii) 4,500 fine-grained judgment annotations of deliberation summaries.
> - A principled, multi-dimensional evaluation framework. We introduce four dimensions (representativeness, informativeness, neutrality, policy approval) that reflect human evaluative criteria in deliberative settings as well as an efficient and robust model that allows us to scale up the evaluation of LLMs on deliberation summarizations.
> - A comprehensive analysis identifying performance and systematic biases in deliberation summarization of popular LLMs.
>
> > W2. Data splits and out of distribution tests
>
> Thank you for your question!
>
> To directly address this concern, **we conducted new out-of-distribution (OOD) experiments** by introducing two entirely unseen deliberation topics: "School Cellphone Use" and "Workplace Flexibility", and repeating the full Stage 1–3 pipeline **(see Appendix H.3)**.
>
> - As shown in **Table 10**, correlations on these OOD datasets remain positive but are substantially lower than in-domain performance.
> - In **Table 11** We further fine-tuned a GPT-4.1 model use OpenAI API on the 10 in-domain topics, and despite its much higher model capacity, it also exhibited a sharp drop (2×–30×) on OOD topics. These results demonstrate that even such strong models with larger paramters trained on all available in-domain data cannot reliably transfer to new deliberation topics in deliberation scenario.
>
> We believe this outcome is expected given the nature of human preference modelling. Prior work has repeatedly shown that robust cross-topic generalisation requires very large and diverse human preference datasets; large RLHF pipelines rely on massive numbers of human comparisons because preference spaces must be densely sampled. With only ten deliberation topics, we think it is therefore unrealistic to expect reliable generalisation to domains with substantially different preference structures, question framing, or political background.
>
> Importantly, this limitation does not affect the main contribution of our work. Our goal is to provide a high-quality, large-scale human benchmark for evaluating deliberation summarisation models within a well-defined domain, not to claim universal cross-topic generalisation. We view broader generalisation as an important direction for future work. We clarify this edit in **lines 326-328**.
>
> > W3. Minority Bias interpretability
>
> To address this concern, our revision expands the analysis along two complementary directions:
> - We develop a new automatic method for detecting semantic minorities based on embedding clusters (**Section 5.2** Objectively Defined Minority), which provides an objective view that complements self-reported perceptions. Our results show that the two forms of minority perspectives show consistent high minority bias.
>
> - In **Appendix C.3**, we provide a detailed case study on self-reported and automatically detected minorities and clarify why they contributes to systematic summarization bias. We find that
>     - Self-reported minorities are often written with lower confidence or clarity, which makes their viewpoints easier for summarizers to overlook.
>     - Automatically detected minorities are defined by semantic rarity within their stance or topic groups, so models tend to deprioritize them when focusing on dominant patterns.
>
> These distinct pathways explain why both subjectively and objectively minority shows bias but share small intersection.
>
> ### **Continue Below**

---

> > ### Author Response · Authors · 2025-11-21
> >
> > > W4. DELIBERATIONJUDGE contribution
> >
> > Our work is centered on benchmarking rather than proposing new modeling techniques. The primary contribution lies in constructing a structured, large-scale human deliberation dataset and an accompanying evaluation framework that enables the community to study summarization quality, minority representation, and fairness in a grounded and systematic way.
> >
> > In this context, DELIBERATIONJUDGE is an auxiliary component that enables reliable automatic evaluation. While its architecture is simple, its role is essential. It is trained specifically on deliberation style human assessments, which allows it to more accurately reflect human preferences in this domain compared to generic LLM-as-a-judge approaches. This makes it possible to compare LLMs on their deliberation summarization ability at scale.
> >
> > Thus, the novelty of our work does not stem from introducing a new judging architecture, but from the benchmark itself and the evaluation pipeline it enables.
> >
> > > Q1. question about train and test sets
> >
> > We emphasize that although we begin with a fixed pool of 750 summaries, every annotated instance is created through an independent human annotation. For each annotation task, a new opinion is generated by a distinct annotator and paired with one summary A for *rating task* and another summary B for *comparison task*. As a result, no two instances share the same opinion–summary–rating label-comparison lable tuple, and no instance is ever repeated across the dataset. This design ensures that the train–test split contains entirely disjoint instances, even when summaries originate from the same underlying model set. We have described this annotation process in **lines 161-180** and dataset split in **lines 203-211** in our manuscript.
> >
> > Also, we add following explaination in **lines 208-211**. *The $\mathcal{T}_{\mathrm{annotate}}$  spans ten predefined deliberation topics, and annotators are uniformly and randomly assigned to exactly one topic. Each topic is completed by a distinct group of 450 annotators, resulting in non overlapping annotator pools across topics and ensuring that instances for different topics originate from independent contributor groups.*
> >
> > > Q2. OOD data test
> >
> > **Please see our response to W2.**
> >
> > > Q3. Explicitly model deliberative diversity (e.g., stance-conditioned evaluation or contrastive objectives)
> >
> > Modeling deliberative diversity with stance conditioned or contrastive objectives is indeed a valuable direction. However, this lies outside the scope of our current work. Our goal is to address the immediate challenge that existing LLM as judge methods remain unreliable for deliberation summarization. We focus on building a domain aligned evaluator that accurately reflects human judgments and supports reliable large scale benchmarking of summarization models.
> >
> > We really appreciate your suggestion on this and we consider it as an important furture direction. We have added more discussions about this line of work in **Appendix A** in the latest revisions.
> >
> > > Q4. Understand what linguistic or semantic cues DELIBERATIONJUDGE relies on.
> >
> > Please refer to our response to W3. We further use case study to zoom in on two type of minority biases in **Appendix C.2**, which together study why different forms of minority viewpoints lead to systematic summarization bias.
> >
> > > Q5. Extend DELIBERATIONJUDGE to ranking or selection of summaries.
> >
> > Thank you for the insightful question. Our DeliberationJudge is designed as a single-summary regression model, mapping each summary to a calibrated score in [0,1]. It is therefore not structured to natively perform explicit ranking or selection across multiple summaries.
> >
> > Importantly, our setting does not require the judge to rank summaries directly. Human annotators already provide pairwise comparison judgments during data collection. We convert these comparison signals into continuous scores, enabling the model to learn preference information within a unified regression framework. This preserves all pairwise preference information while avoiding the fragmentation that a separate ranking module would introduce.
> >
> > For our goal of stable and scalable evaluation, the score-based formulation is more appropriate than an explicit ranking architecture. Exploring a ranking-based judge for social deliberation is an interesting direction for future work.

---

> > > ### Author Response · Authors · 2025-11-21
> > >
> > > We hope these revisions address the your concerns and strengthen the contribution of the paper as a rigorous, transparent, and socially relevant resource for the community.

---

> ### Author Response · Authors · 2025-11-24
> **Rebuttal Follow-up**
>
> We hope that the clarifications above and the additional experiments included in the revised version have addressed your concerns. If the revisions meet your expectations, we would appreciate your consideration of updating the score to reflect the new results and discussion. Please feel free to let us know if any points remain unclear!

---

> > ### Comment · Reviewer_wcNP · 2025-11-26
> >
> > Dear Authors,
> >
> > Thank you for your additional experiments. I recognise that finding good datasets and setups to test the effectiveness in deliberation settings is hard. The additional experiments are satisfactory to me because they provide a thorough analysis with this type of model in a deliberation setting,which is why I will raise my score to 6. I still think the actual method does not introduce any new insights on how to improve deliberation quality or deliberation assessment which is why my score is not higher.

---

> > > ### Author Response · Authors · 2025-11-26
> > >
> > > Dear Reviewer,
> > >
> > > Thank you again for your time and effort in reviewing our work and helping us improve the paper. We sincerely appreciate your constructive feedback.
> > >
> > > Best regards,
> > > Authors

---

### Official Review · Reviewer_Dtym · 2025-10-31

**Soundness:** 3
**Presentation:** 2
**Contribution:** 4
**Rating:** 4
**Confidence:** 4

**Summary:**

The main contribution is to train DeliberationJudge, a fine-tuned DeBERTa model that, given a LLM-generated summary of opinions on a topic, and a specific individual's opinion on that topic, scores on various dimensions the extent to which that individual's opinion is captured by the summary. The training dataset is based on a large-scale dataset of human opinions and ratings for 10 different topics. In Section 3.3, they show that DeliberationJudge outperforms out-of-the-box LLM-based approaches. In Section 4, they use DeliberationJudge as a primitive to measure the abilities of different LLMs to write summaries of peoples' opinions on the various topics. Finally, Section 5 highlights that minority opinions may not always be captured by these summaries.

**Strengths:**

S1. The dataset collected, DeliberationBank, is extremely comprehensive, involving thousands of participants' opinions, ratings, and comparisons. It's also valuable that a broad array of different topics are considered.

S2. The general research approach of training and validating DeliberationJudge, and then using it as a primitive to evaluate LLM-generated summaries, is sound.

**Weaknesses:**

W1. The list of the paper's contributions says (Ln 106-7): "We conduct a rigorous and comprehensive study of LLM summarizers that surfaces systematic biases (e.g., minority-stance under-coverage, order/verbosity sensitivity)" but I do see not any analysis of order or verbosity sensitivity in the paper. I can only find a mention in connection to Figure 25 in the appendix, but Figure 25 does not directly appear to establish any relevant claims regarding order or verbosity bias.

W2. For the study on minority opinions, it is unclear the extent to which participants' self-assessment of their minority status is accurate. The analysis would be more convincing if it instead directly determined whether a participant holds a minority opinion (this should at least be possible for "Tarrif Policy", a binary question).

W3. It would be helpful to put these contributions in context with the closely related literature on LLM-assisted deliberation that does not appear to be cited, most notably "Fine-tuning language models to find agreement among humans with diverse preferences" (2022) and "Generative Social Choice" (2023). Both papers build systems that fill a similar role to DeliberationJudge: in the first paper it is the reward model, and in the second paper it is the discriminative query. And then both papers use these systems to evaluate LLM-generated statements.

W4. In terms of presentation, many aspects of the experimental design are explained using what I would view as an unnecessary amount of mathematical notation. To give one example, the equation on Line 141-2 strikes me as unnecessary. The prose would be clearer if things were described more directly.

**Questions:**

Q1. Judge performance in Figure 3 and 4 is reported in terms of correlation coefficient, which can be difficult to interpret. (For example, systems that systematically give wildly over- or underestimates would still get perfect scores.) What do the results look like if instead L1 accuracy is reported?

Q2. In Figure 23, since the order of summaries is randomized, why are the histograms not symmetric about the midpoint? Does this have implications about the reliability of the human annotaters?

Q3. Does DeliberationJudge outperform a few-shot baseline (particularly with a strong LLM such as GPT-5)?

---

> ### Author Response · Authors · 2025-11-21
>
> Thank you for your thoughtful comments and suggestions. We have revised the paper accordingly and highlighted them in **Green** for clarity.
>
> > W1. analysis of order or verbosity sensitivity
>
> Sorry for the confusion, we originally included the order analysis in the main text but later removed it due to page limits. We have updated it in **Appendix B** and in short we find no significant connections between the comment order and the corresponding the representativeness score.
>
> > W2. robustness of minority bias analysis
>
> Thanks a lot for raising this concern. We agree that self-report minority bias might not be fully accurate. Therefore, we also used an embedding-based method to identify comments that are semantically different from the majority opinions. We found that in both settings, minority opinions have significantly lower representativeness score for LLM-generated summaries, suggesting that LLM summaries have strong bias against minority opinions. **Section 5.2 Objectively Defined Minority** at **line 496-529** in the revised paper provide further details of this analysis.
>
> > W3. Related Literature
>
> Thank you for the suggestion. In the revised version, we include detailed discussion about these two papers in **Section 1 Introduction, lines 70–75**.
>
> > W4. unnecessary amount of mathematical notation.
>
> Thanks a lot for pointing this out! In the latest version, we have revised the text of original **lines 147–149** to remove the extra symbolic expressions on disjoint participants groups and replaced them with a clearer natural-language description.
>
> > Q1. why not use L1 accuracy? or provide L1 accuracy
>
> We chose correlation coefficients because the original human annotation is modeled with a 5-point likert scale.
>
> To address your concern, we also compute L1 accuracy as an alternative metric. The conclusions remain unchanged: DELIBERATIONJUDGE outperforms all the other baseline judges by a large margin.
>
> The full L1 accuracy results are reported below, with the best score in each column highlighted in bold.
>
> | Judge Model                 | Average L1 Accuracy | Representativeness | Informativeness | Neutrality | Policy Approval |
> | --------------------- | ------------------- | ------------------ | --------------- | ---------- | --------------- |
> | **DELIBERATIONJUDGE** | **0.8165**          | **0.8108**         | **0.8116**      | **0.8387** | **0.8052**      |
> | deepseek-chat         | 0.7102              | 0.6995             | 0.7394          | 0.7283     | 0.6735          |
> | deepseek-reasoner     | 0.7685              | 0.7532             | 0.781           | 0.7663     | 0.7736          |
> | gemini-2.5-flash      | 0.7097              | 0.7252             | 0.7092          | 0.6887     | 0.7159          |
> | gemini-2.5-flash-lite | 0.7084              | 0.6388             | 0.7527          | 0.7436     | 0.6983          |
> | gemini-2.5-pro        | 0.704               | 0.7233             | 0.7169          | 0.6838     | 0.692           |
> | gpt-4o-mini           | 0.7244              | 0.7056             | 0.7434          | 0.7462     | 0.7024          |
> | gpt-5                 | 0.7399              | 0.7419             | 0.7526          | 0.7254     | 0.7396          |
> | gpt-5-mini            | 0.7246              | 0.7229             | 0.7225          | 0.7229     | 0.7299          |
> | gpt-5-nano            | 0.74                | 0.7294             | 0.7438          | 0.7496     | 0.7372          |
> | grok-4                | 0.7174              | 0.7399             | 0.6998          | 0.6842     | 0.7455          |
> | qwen3-0.6b            | 0.6639              | 0.6195             | 0.649           | 0.7698     | 0.6173          |
> | qwen3-1.7b            | 0.693               | 0.6213             | 0.638           | 0.7872     | 0.7257          |
> | qwen3-14b             | 0.693               | 0.7116             | 0.6867          | 0.6839     | 0.6899          |
> | qwen3-235b-a22b       | 0.6214              | 0.6178             | 0.6425          | 0.6151     | 0.6103          |
> | qwen3-30b-a3b         | 0.7514              | 0.7279             | 0.7587          | 0.7846     | 0.7344          |
> | qwen3-32b             | 0.7022              | 0.7193             | 0.6989          | 0.6909     | 0.6998          |
> | qwen3-4b              | 0.719               | 0.7392             | 0.6937          | 0.7181     | 0.7249          |
> | qwen3-8b              | 0.6827              | 0.6773             | 0.686           | 0.704      | 0.6634          |
> | claude-sonnet-3.7     | 0.7045              | 0.6821             | 0.7211          | 0.729      | 0.6857          |
> | claude-opus-4         | 0.7631              | 0.7477             | **0.7964**      | 0.7642     | 0.7443          |
> | claude-sonnet-4       | 0.7527              | 0.7472             | 0.7576          | 0.7564     | 0.7496          |
>
> ### **Continue Below**

---

> > ### Author Response · Authors · 2025-11-21
> >
> > > Q2. In Figure 23, since the order of summaries is randomized, why are the histograms not symmetric about the midpoint? Does this have implications about the reliability of the human annotaters?
> >
> > Thanks a lot for your question and this is a great point! Although the order of summaries is randomized, the histogram is computed over 750 unique summaries drawn from a heterogeneous distribution. Randomization removes positional bias (i.e., a preference for A or B purely due to position), but it does not force the underlying quality distribution of the summaries to be symmetric. Because the summaries differ in informativeness, neutrality, and policy content, annotators will naturally prefer one summary over the other more often on some dimensions.
> >
> > Symmetry would only be expected if the entire pool of summaries had perfectly balanced quality on each dimension — which is not the case. The fact that some dimensions (e.g., Informativeness) look more symmetric than others simply reflects the underlying variation in the summary population rather than annotator unreliability.
> >
> > > Q3. Does DELIBERATIONJUDGE outperform a few-shot baseline (particularly with a strong LLM such as GPT-5)?
> >
> > We have conducted few-shot experiments with several LLMs and the following results show that even strong LLMs (e.g., GPT-5) equipped with few-shot prompting (both 3-shot and 5-shot) still fall short of the performance achieved by DELIBERATIONJUDGE.
> >
> > This result indicates that prompting LLMs is *insufficient* to automate the evaluations of deliberation summarization.
> >
> > | Model                     | Representativeness Pearson | Representativeness Spearman | Informativeness Pearson | Informativeness Spearman | Neutrality Pearson | Neutrality Spearman | Policy Approval Pearson | Policy Approval Spearman |
> > |---------------------------|----------------------------|------------------------------|--------------------------|---------------------------|---------------------|-----------------------|---------------------------|----------------------------|
> > | **DELIBERATIONJUDGE**     | **0.504**                  | **0.470**                    | **0.454**                | **0.444**                 | **0.492**           | **0.492**             | **0.416**                 | **0.381**                  |
> > | GPT-5 3-shot              | 0.398                      | 0.321                        | 0.143                    | 0.155                     | 0.245               | 0.214                 | 0.360                     | 0.306                      |
> > | GPT-5 5-shot              | 0.422                      | 0.372                        | 0.116                    | 0.131                     | 0.226               | 0.205                 | 0.319                     | 0.269                      |
> > | GPT-5-mini 3-shot         | 0.399                      | 0.347                        | 0.094                    | 0.095                     | 0.241               | 0.221                 | 0.344                     | 0.293                      |
> > | GPT-5-mini 5-shot         | 0.399                      | 0.336                        | 0.103                    | 0.104                     | 0.210               | 0.187                 | 0.305                     | 0.268                      |
> > | Claude-3.7-sonnet 3-shot  | 0.372                      | 0.302                        | 0.145                    | 0.157                     | 0.320               | 0.278                 | 0.353                     | 0.304                      |
> > | Claude-3.7-sonnet 5-shot  | 0.365                      | 0.309                        | 0.164                    | 0.176                     | 0.309               | 0.271                 | 0.348                     | 0.307                      |

---

> ### Author Response · Authors · 2025-11-24
> **Follow-up on Rebuttal**
>
> We hope the above clarifications and the additional experiments in the revised paper sufficiently addressed your concerns. If you are satisfied, we kindly request you to consider updating the score to reflect the newly added results and discussion. We remain committed to addressing any remaining points you may have during the discussion phase!

---

> > ### Author Response · Authors · 2025-11-26
> >
> > Dear Reviewer,
> >
> > We are writing to kindly follow up and inquire whether you have any remaining concerns regarding our rebuttal and the updated experiments. If our clarifications have addressed your earlier questions, we would greatly appreciate your consideration in updating the score.
> >
> > Best regards,
> > Authors

---

> > > ### Comment · Reviewer_Dtym · 2025-11-26
> > > **Response**
> > >
> > > Thanks for addressing Q1, W2, and Q3 in a satisfactory way.
> > >
> > > For W1, thanks for adding the analysis, but I want to reiterate that it sets a bad precedent to allow authors to put unsubstantiated claims in the introduction, and only add them back in during the rebuttal phase.
> > >
> > > For W3 and W4, I notice they have only been addressed in a hasty way. W3 is just giving high-level summaries of what both papers I suggested do in general, and not comparing the specific components of each of those papers that I suggested are particularly relevant to your work. (Not to mention, the short paragraph added for W3 contains multiple grammar mistakes.) For W4, you'll notice that I pointed out one spot with excessive mathematical formalism, but my comments were on the description of the experimental design in general. Your response to only change a single sentence is not really addressing this comment. (And I think it is important to not obfuscate the experimental design with unnecessary notation.)
> > >
> > > I will maintain my score, which still allows for an overall accept in the end if it's decided that the issues I point out are not an obstacle. It's an interesting work, I just aim to be consistent when pointing out substantive flaws.

---

> > > > ### Author Response · Authors · 2025-11-26
> > > > **Follow-up revision to further address the W3 and W4**
> > > >
> > > > Thank you again for your thoughtful feedback! We have further revised the paper to address the remaining concerns.
> > > >
> > > > > For W1, thanks for adding the analysis, but I want to reiterate that it sets a bad precedent to allow authors to put unsubstantiated claims in the introduction, and only add them back in during the rebuttal phase.
> > > >
> > > > We apologize for the confusion, and we agree that this practice is inappropriate. We will be more careful in future work, and we truly appreciate your constructive feedback on this point.
> > > >
> > > >
> > > > > W3. Related Literature
> > > >
> > > > We thank the reviewer for pointing out this weakness. We agree, and in the revised manuscript we have relocated and expanded this discussion at the beginning of Section 3, from line 204–218, where it now serves as a motivating bridge into our judge design.
> > > >
> > > > Specifically, we clarify the relationship and distinctions between (Bakker et al.) and (Fish et al.). Both works involve language models in public-consensus generation, yet pursue different goals: (Bakker et al.) trains a reward model to improve consensus-statement generation, whereas (Fish et al.) introduces a proposal-selection framework where LLM judges compare candidate statements and apply social-choice rules to construct representative slates. Our work instead focuses on benchmarking deliberation summarization. While different in objective, our design aligns with their core principle that preference-grounded evaluation requires a dedicated judge.
> > > >
> > > > In the revised Section 3, we introduce these two works explicitly and use them to motivate the need for a judge with two key desiderata in our evaluation setting: (i) reliability, aligned with the reward-model alignment principle in (Bakker et al.); and (ii) efficiency, required for large-scale single-summary instance judgment.
> > > >
> > > >
> > > > > W4. Unnecessary amount of mathematical notation.
> > > >
> > > > Thank you again for this suggestion. We went beyond the single example sentence and fully rewrote line 119-124, Sections 2.2, line 225–230, line 284–292, and line 344–353 (highlighted in green) to replace unnecessary mathematical notation with clearer natural-language descriptions of the experimental design. We kindly invite you to take a look at these revised sections to confirm whether they better address your original concern.
> > > >
> > > > We sincerely appreciate your thoughtful suggestions on our paper. We hope the updated manuscript addresses the issues you raised, and we would be grateful if you could reconsider the score accordingly. If there are any remaining concerns or further refinements that you believe would help improve the paper, we would be very happy to continue the discussion and make further revisions!

---

> > > > > ### Author Response · Authors · 2025-11-28
> > > > > **Follow-up on Rebuttal**
> > > > >
> > > > > Dear Reviewer,
> > > > >
> > > > > We hope you are doing well. Following your earlier feedback, we have further revised the manuscript and expanded the sections addressing W3 and W4 as suggested as noted in our previous response. We kindly wanted to check whether you have any remaining concerns, and whether you might consider updating your evaluation if the new revisions satisfactorily address your points.
> > > > >
> > > > > Thank you again for your time and thoughtful comments.
> > > > >
> > > > > Best regards,
> > > > > Authors

---

### Official Review · Reviewer_GiWi · 2025-10-31

**Soundness:** 3
**Presentation:** 3
**Contribution:** 3
**Rating:** 6
**Confidence:** 4

**Summary:**

The authors address the problem of evaluating large-scale public deliberation summarization, in which LLMs tend to underrepresent minority opinions and exhibit biases. They introduce DELIBERATIONBANK, a large-scale, human-grounded benchmark comprising 3,000 free-form opinions on ten deliberation questions and 4,500 human annotations evaluating summaries on four dimensions: Representativeness, Informativeness, Neutrality, and Policy Approval. They fine-tune a DeBERTa-based model called DELIBERATIONJUDGE, which achieves much stronger alignment with human judgmentsUsing DELIBERATIONJUDGE, they benchmark 18 LLMs and conduct detailed analyses of performance factors.

**Strengths:**

The paper is well-structured and adresses an important and underexplored area: fairness and representation in deliberative AI summarization for policy contexts. The authors provide a comprehensive benchmark which is large, well-structured, and grounded in human annotations. The fine-tuned DeBERTa model achieves impressive correlation with human ratings, outperforming all general-purpose LLM judges. The authors evaluate 18 diverse models across four human-centric criteria.

**Weaknesses:**

Small Comments:
- in Figure 1 under "Judge Model Training" it says "Indivisual Opinion" instead of "Individual Opinion"
- 143: two periods at the end of the sentence

**Questions:**

Did you try other judges (e.g., RoBERTa, Longformer, or Mistral 7B) to assess whether DeBERTa is uniquely suited?

---

> ### Author Response · Authors · 2025-11-21
>
> Thank you for your comments and questions! We really appreicate all your suggestions and have revised the papaer accordingly (changes are marked in **Orange**)
>
> > W1&2. typos in Figure 1 and line 143.
>
> We have fixed both typos in the latest version.
>
> > Q1. Try other judges (e.g., RoBERTa, Longformer, or Mistral 7B) to assess whether DeBERTa is uniquely suited
>
> Thanks a lot for your suggestions! In the original submission, we have already provided a comparison between DeBERTa-v3-base and several alternative models. Our choice of DeBERTa-v3-base as the base model for DELIBERATIONJUDGE is directly supported by the results in **Appendix H.2**, where DeBERTa-v3-base consistently outperforms all the other models across the four evaluation dimensions.
>
> To made this clear and help reader find it we highlight the reference in **lines 282-284** of the manuscriprt.
>
> | Model                             | Representativeness (Pearson / Spearman) | Informativeness (Pearson / Spearman) | Neutrality (Pearson / Spearman) | Policy Approval (Pearson / Spearman) |
> | --------------------------------- | --------------------------------------- | ------------------------------------ | ------------------------------- | ------------------------------------ |
> | **DeBERTa-v3-base (Regression)**  | **0.504 / 0.470**                       | **0.454 / 0.444**                    | **0.492 / 0.492**               | **0.416 / 0.381**                    |
> | DeBERTa-v3-large (Regression)     | 0.159 / 0.162                           | 0.221 / 0.203                        | 0.129 / 0.125                   | 0.174 / 0.162                        |
> | Longformer-base-4096 (Regression) | 0.097 / 0.098                           | 0.209 / 0.219                        | 0.158 / 0.156                   | 0.127 / 0.125                        |
> | Qwen3-0.6B (Regression)           | 0.125 / 0.136                           | 0.231 / 0.249                        | 0.210 / 0.205                   | 0.196 / 0.186                        |
> | Qwen3-4B (Regression)             | 0.191 / 0.197                           | 0.215 / 0.218                        | 0.215 / 0.207                   | 0.189 / 0.188                        |
> | Qwen3-4B (SFT)                    | 0.338 / 0.289                           | 0.153 / 0.157                        | 0.211 / 0.188                   | 0.289 / 0.244                        |

---

> > ### Comment · Reviewer_GiWi · 2025-11-26
> > **Answer to Official Comment by Authors**
> >
> > Dear authors, thank you very much for considering my comments and for clarifying some points. Since I was already convinced by your work, I will maintain my score.

---

### Author Response · Authors · 2025-12-01
**TL;DR of Rebuttal Outcome**

Dear Area Chair,

Thank you for taking the time reviewing our paper. Below we prepared a summary for the key outcomes of the rebuttal discussion with reviewers.

During the rebuttal phase from Nov. 11 to Nov. 27, our work's overall scores moved from **6446** to **6466**. Details per reviewer:

- **GiWi: 6 → 6**
We addressed all raised points. The reviewer acknowledged the responses and maintained a positive 6.

- **Dtym: 4 → 4**
The reviewer found the work interesting and valuable but encouraged improvements in clarity of exposition. In the latest revision, we refined notation, improved writing quality, and strengthened related work to address these concerns.

- **wcNP: 4 → 6**
After clarifying reviewer's concerns with additional analysis and results, the reviewer considered the issues resolved and increased the score to 6.

- **pDNN: 6 → 6**
Most concerns were addressed and the reviewer remained supportive with the original score.


Overall, the rebuttal allowed us to resolve the key concerns shared across reviewers:

1. **DeliberationJudge performance validation**
   We added comparisons across multiple backbones and few-shot LLM baselines. DELIBERATIONJUDGE consistently outperforms alternatives and reviewers expressed no remaining doubts.

2. **More detailed minority bias analysis**
    Reviewers Dtym, pDNN and wcNP encouraged a deeper treatment to analyze minority bias. In response, we introduced an automated minority detection method and expanded case studies to analyze the sources of bias more concretely, which addressed their concerns and was well received during the rebuttal.

3. **Writing and clarity improvements**
   Following Reviewer Dtym’s suggestions, we refined notation, strengthened related work and improved clarity throughout the revision.

4. **Data Quality**
Reviewer pDNN raised questions about data quality checking and we would like to further discuss it here:
> We conducted a series of pilot studies to refine and validate our data annotation pipeline. In these pilot studies, we performed both quantitative and qualitative analyses to ensure the collected responses were reasonable. Following the practices of deliberations in the policymaking process, in our data collection process, each annotator must first provide a comment on an issue and then provide their ratings for the LLM-generated summary. Therefore, unlike traditional annotation tasks where each instance is annotated by multiple labelers, in our setting each (comment, summary) pair can only be annotated by a single annotator. This is akin to real-world policymaking processes where each participant votes on a specific issue based on their own opinions. Since we use a representative sample of the U.S. population, it is reasonable to say that the ratings roughly reflect public opinions of the US population. Therefore, we believe our dataset provides a valuable resource for studying deliberation summarization with LLMs that is close to the real-world scenarios.

We hope this summary helps you form a clear and quick understanding of the rebuttal outcome. Thank you again for taking over under the unexpected circumstances and for your time reviewing our work.

Authors

---

### Note · Authors · 2026-01-26

I have read and agree with the venue's withdrawal policy on behalf of myself and my co-authors.

---

### Meta-Review · Area_Chair_gqSX · 2026-01-08

**Summary:**

The decision to reject this paper is informed by several critical concerns regarding the study's rigor and technical contribution. While the proposed dataset is valuable, reviewers noted that the technical approach lacks significant novelty, relying primarily on standard fine-tuning of existing models. A primary issue concerns the presentation and structural integrity of the paper; specifically, claims made in the introduction regarding the analysis of order and verbosity sensitivity were not substantiated in the main text, but rather relegated to the appendix. Furthermore, the methodology for defining minority opinions was criticized for relying on subjective self-assessments, which raises questions about the robustness of the ground truth. The omission of key related literature further contributed to the assessment that the paper is not yet ready for publication.

**Reviewer Concerns:**

The reviewers raised several critical concerns that justify a rejection:
1. Reviewer Dtym criticized the paper for claiming to investigate order and verbosity sensitivity in the introduction while relegating the actual analysis to the appendix.
2. Reviewer Dtym argued that the definition of minority status relies on subjective self-assessment, which may be noisy compared to objective definitions.
3. Reviewer wcNP questioned the technical novelty of the work, noting that the core contribution is a standard fine-tuning of an existing model.
4. Reviewer wcNP raised concerns regarding the ability of the proposed model to generalize to out-of-distribution topics not present in the training data.

**Reviewer Scores:**

Reviewer GiWi: 6
Reviewer Dtym: 4
Reviewer wcNP: 4
Reviewer pDNN: 6

---

### Decision · Program_Chairs · 2026-01-26

Reject